# Underestimated Privacy Risks for Minority Populations in Large Language Model Unlearning

## Abstract

Large Language Models (LLMs) are trained on extensive datasets that often contain sensitive, human-generated information, raising significant concerns about privacy breaches. While certified unlearning approaches offer strong privacy guarantees, they rely on restrictive model assumptions that are not applicable to LLMs. As a result, various unlearning heuristics have been proposed, with the associated privacy risks assessed only empirically. The standard evaluation pipelines typically randomly select data for removal from the training set, apply unlearning techniques, and use membership inference attacks (MIAs) to compare the unlearned models against models retrained without the to-be-unlearned data. However, since every data point is subject to the right to be forgotten, unlearning should be considered in the worst-case scenario from the privacy perspective. Prior work shows that data outliers may exhibit higher memorization effects. Intuitively, they are harder to be unlearn and thus the privacy risk of unlearning them is overlooked and underestimated in the current evaluation. In this paper, we leverage minority data to identify such a critical flaw in previously widely adopted evaluations. We substantiate this claim through carefully designed experiments, including unlearning canaries related to minority groups, inspired by privacy auditing literature. Using personally identifiable information (PII) as a representative minority identifier, we demonstrate that minority groups experience at least 20% more privacy leakage in most cases across six unlearning approaches, three MIAs, three benchmark datasets, and two LLMs of different scales. Given that the right to be forgotten should be upheld for every individual, we advocate for a more rigorous evaluation of LLM unlearning methods. Our minority-aware evaluation framework represents an initial step toward ensuring more equitable and thorough assessments of LLM unlearning efficacy.

## 1 Introduction

Large Language Models (LLMs) are trained on vast and diverse datasets, often sourced from public content on the web, much of which is generated by humans (Touvron et al., 2023; Ouyang et al., 2022). This practice raises significant ethical concerns, particularly when the data includes sensitive information, leading to potential privacy violations. Individuals whose data has been used may seek to exercise their "right to be forgotten", a protection guaranteed by regulations such as the General Data Protection Regulation (GDPR) (Krzysztofek, 2018).

The ideal approach to fulfilling such a request is to retrain the LLM from scratch, excluding the data to be removed. However, this solution is prohibitively expensive and impractical for large-scale models. To address this, the concept of *machine unlearning* has emerged as a promising alternative. Machine unlearning seeks to efficiently modify the LLM so that it becomes statistically indistinguishable from a model retrained from scratch, without the data subject to removal. In this way, no adversary could confidently determine whether a model has undergone a proper unlearning process or been retrained, ensuring compliance with the "right to be forgotten".

Unfortunately, it remains an open problem to enforce the formal unlearning guarantee for deep neural networks and LLMs. Despite recent progress in theoretical unlearning literature (Guo et al., 2020; Sekhari et al., 2021; Neel et al., 2021; Ullah et al., 2021; Chien et al., 2023; Ullah & Arora, 2023;

Chien et al., 2024a;b), they are still far from ready to be applied to deep neural networks and LLMs due to their restrictive assumptions. Meanwhile, researchers have developed efficient unlearning heuristics and verified the efficacy of unlearning empirically (Golatkar et al., 2020a;b; Graves et al., 2021; Liu et al., 2024a;c; Yao et al., 2024). Recent advancements in LLM unlearning often measures the unlearning performance by comparing the performance of unlearned models against models retrained from scratch (Pawelczyk et al., 2024a). A common approach to empirically evaluating unlearning efficacy is to employ membership inference attacks (MIAs) (Shokri et al., 2017), where an attacker attempts to identify if a model is from retraining or not. For instance, MUSE (Shi et al., 2024b) is a representative benchmark study that leverages the ROC-AUC metric from MIAs to quantify privacy leakage.

We identify a critical pitfall in the aforementioned LLM unlearning efficacy evaluation pipeline. Literature indicates that memorization levels in LLMs vary significantly across individual training samples (Feldman & Zhang, 2020; Carlini et al., 2022). However, the current LLM unlearning efficacy evaluation only captures an 'average-case' performance, where removed data is randomly sampled from the training set. This approach neglects the privacy risk of data that are hard to unlearn which does not account for situations where privacy must be rigorously protected in the worst-case scenario (Steinke & Ullman, 2020; Aerni et al., 2024). It neglects the principle that every individual's right to be forgotten should be upheld equally, where the privacy risk of data from minority groups is overlooked. They are often considered as outliers and may be more resistant to unlearning efforts due to the aforementioned stronger memorization effect. Therefore, standard unlearning evaluation significantly underestimates privacy risks associated with these groups within the training set, which also misses the broader social responsibilities.

To verify our claim, we first conduct a careful synthetic experiment of unlearning injected canaries pertaining to minority groups, which is motivated by the privacy auditing literature (Jagielski et al., 2020; Steinke et al., 2024). We choose Personally Identifiable Information (PII) as a representative minority identifier, while our idea extends to broader cases. We show that minorities suffer from at least 20% more of privacy leakage in most cases across combinations of six unlearning approaches, three variants of MIAs, three benchmark datasets, and two LLMs of different scales. It highlights the prevalence of the claimed issue in practical settings. Our results call for a more careful LLM unlearning efficacy evaluation, particularly in regard to privacy risk for minority groups. We propose a minority-aware LLM unlearning evaluation protocol (Figure 1) as an initial step toward this goal. We further benchmark existing unlearning approaches and investigate the effect of the forget set size as well as unlearning complexity with our minority-aware evaluation. This provides a more holistic understanding of different LLM unlearning approaches for practitioners. Importantly, we observe that Langevin Unlearning—the only approach incorporating noise—achieves a favorable privacy-utility trade-off. This suggests that incorporating noise may play a crucial role in effective unlearning.

## 2 RELATED WORK

Privacy auditing is a fundamental yet challenging aspect of LLM unlearning due to the difficulty of distinguishing training samples effectively (Duan et al., 2024). Various privacy-related metrics, such as exposure (Carlini et al., 2019), mean reciprocal rank (Wu et al., 2023), extraction likelihood (Jang et al., 2022), and truth ratio (Maini et al., 2024), have been proposed to probe privacy leakage. Among these, MIAs remain one of the most crucial tools for evaluating machine unlearning methods (Liu et al., 2024b). Standard MIAs typically involve training numerous shadow models independently to empirically approximate the distribution (Carlini et al., 2022). This approach has also been adopted for LLM unlearning, as seen in the NeurIPS 2023 Machine Unlearning Challenge[1] (which compares the point-wise output distributions of multiple unlearned and retrained models to perform MIAs) and Kurmanji et al. (2024); Pawelczyk et al. (2024b). Hayes et al. (2024) further highlights the limitations of average-case evaluations and proposes the specialized MIA method for unlearning evaluation to date. Their work focuses on unlearning a randomly selected subset of training data with a sample-dependent MIA. By contrast, our work targets unlearning cases involving minority populations within the training data, using a fixed MIA for evaluation.

A major downside of MIA approaches involving shadow models is their computational expense, as they require training a large number of LLMs independently (Liu et al., 2024b). To address this

---

[1]`https://unlearning-challenge.github.io/assets/data/Machine_Unlearning_Metric.pdf`

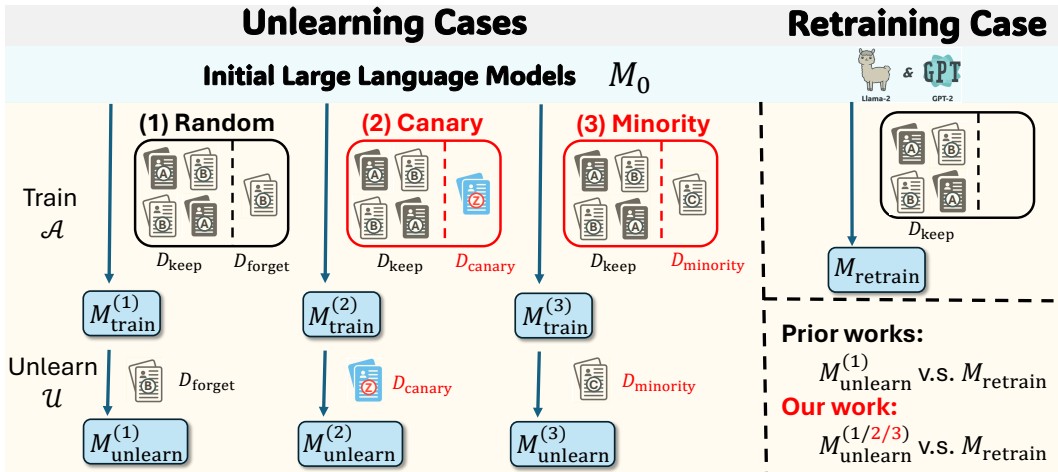

Figure 1: Illustration of the existing LLM unlearning evaluation pipeline with our proposed approaches (highlighted in red). Standard LLM unlearning evaluation typically involves randomly sampling data for removal from the training set (Case 1), which may underestimate privacy leakage for minority groups. In contrast, we design experiments to assess unlearning efficacy by removing canaries related to minority groups (Case 2) and by directly removing data from minority groups (Case 3). Our approach provides a more comprehensive, minority-aware evaluation by considering the worst result across the three settings.

downside, another line of research compares the outputs of models using different statistical metrics without requiring shadow models (Zhang et al., 2024; Liu et al., 2024a;c; Yao et al., 2024; Li et al., 2024), making these methods more computationally feasible Maini et al. (2024). For instance, Shi et al. (2024b) measure privacy risk through the normalized AUC difference between unlearned and retrained models, using MIAs such as Min-K% (Shi et al., 2024a). However, these works typically select the forget set randomly from the training set, corresponding to an average-case evaluation. Our study highlights a critical limitation of this approach: the privacy risks of minority populations within the training set are severely underestimated because minority data are less likely to be selected in the unlearning evaluation pipeline. By focusing on minority-aware scenarios, our work provides a more nuanced perspective on unlearning evaluation and privacy risks.

## 3 PRELIMINARIES

Machine unlearning (Cao & Yang, 2015; Bourtoule et al., 2021) has emerged as an important direction in trustworthy language models. It was initially motivated by privacy due to "the right to be forgotten" from GDPR and later on extended to other legal and ethical concerns, including copyright (Yao et al., 2024), biased or outdated information mitigation (Liu et al., 2024b), hallucination removal (Yao et al., 2023), entity forgetting (Maini et al., 2024) and data poisoning removal (Pawelczyk et al., 2024a). In this work, we focus on the privacy aspect of the problem, albeit our methodology extends to other cases whenever the indistinguishability to the retrained model is an appropriate metric.

We briefly state the generic machine unlearning setting for privacy. Assume a training dataset $D_{\text{train}}$ and a holdout test set $D_{\text{test}}$ are given. Let $M_{\text{learn}} \leftarrow \mathcal{A}(M_0, D_{\text{train}})$ be the language model train on $D_{\text{train}}$ starting from an initial model $M_0$ via the training algorithm $\mathcal{A}$, which may be either a pre-trained language model or random initialization. Once the model is trained, we receive data removal requests that partition the training set $D_{\text{train}} = D_{\text{forget}} \cup D_{\text{keep}}$ into a subset to be forgotten later $D_{\text{forget}}$ and a keep set $D_{\text{keep}}$[2]. An unlearning algorithm $\mathcal{U}$ takes $M_{\text{learn}}$, $D_{\text{forget}}$ and $D_{\text{train}}$ as input to return an updated model $M_{\text{unlearn}} \leftarrow \mathcal{U}(M_{\text{learn}}, D_{\text{forget}}, D_{\text{train}})$. It is worth noting that $M_{\text{unlearn}}$ depends on the choice of $D_{\text{forget}}$. The gold standard to adhere "the right to be forgotten" is retraining without $D_{\text{forget}}$, namely $M_{\text{retrain}} \leftarrow \mathcal{A}(M_0, D_{\text{train}} \setminus D_{\text{forget}})$. We say $\mathcal{U}$ achieves good unlearning efficacy if $M_{\text{unlearn}}$ and $M_{\text{retrain}}$ are indistinguishable in their behavior $m(M_{\text{unlearn}}, D) \approx m(M_{\text{retrain}}, D)$ on any corpus

---

[2]We use the term *keep set* instead of the more commonly used term *retain set* in the unlearning literature to prevent confusion between the terms "retain" and "retrain".

$D$, where $m$ is any evaluation metric (Shi et al., 2024b). Since both $M_{\text{unlearn}}, M_{\text{retrain}}$ depends on the choice $D_{\text{forget}}$, such approximation should be taken over the worst case ideally.

### 3.1 Efficient Membership Inference Attacks for LLM Unlearning

As discussed in the previous section, the effectiveness of unlearning methods can be measured by their indistinguishability in behavior compared to a retrained model. Membership Inference Attack (MIA) is often leveraged to determine whether a specific sample is part of the training set and is widely applied to audit training data privacy leakage. Therefore, to evaluate the efficacy of an unlearning approach, we consider the **PrivLeak (PL)** metric (Shi et al., 2024b) defined as follows:

$$\textbf{PrivLeak (PL)} = \frac{\text{AUC}(M_{\text{unlearn}}; D_{\text{forget}}, D_{\text{test}}) - \text{AUC}(M_{\text{retrain}}; D_{\text{forget}}, D_{\text{test}})}{\text{AUC}(M_{\text{retrain}}; D_{\text{forget}}, D_{\text{test}})}, \tag{1}$$

where AUC is the AUC-ROC score of an MIA Ye et al. (2022) that tries to discriminate samples from $D_{\text{forget}}$ and $D_{\text{test}}$ based on the output statistics (e.g. loss) of a given model $M$. By normalizing the difference in AUC scores between $M_{\text{unlearn}}$ and $M_{\text{retrain}}$ using the AUC of $M_{\text{retrain}}$, the metric accounts for the inherent difficulty of distinguishing the forget and test sets. Note that for an effective unlearning method, the metric should be around zero since the behavior of $M_{\text{unlearn}}, M_{\text{retrain}}$ are indistinguishable. A larger magnitude of the PL metric implies a greater amount of privacy information that has been leaked under the tested MIA. A positive value indicates that the sample has not been fully forgotten, as the attacker has a higher AUC for $M_{\text{unlearn}}$ than $M_{\text{retrain}}$. Conversely, a negative metric value suggests over-forgetting, which still indicates that $M_{\text{unlearn}}$ differs from $M_{\text{retrain}}$ and thus may cause privacy breaches. Finally, note that an effective unlearning solution should lead to a small PL metric for *any* choices of MIA. In this work, we consider three popular efficient MIAs and report the corresponding PL metric simultaneously.

- **lossMIA** (Yeom et al., 2018): Determine membership of a sample based on its loss $\ell(M; x)$.

- **zlibMIA** (Carlini et al., 2021): Determines membership of a sample based on the sample loss normalized by its zlib compression size, $\ell(M; x)/\text{zlib}(x)$.

- **Min-K%** (Shi et al., 2023): Selects the lowest $K\%$ of token likelihoods and leverages the corresponding negative log-likelihood for membership inference.

## 4 The Underestimated Privacy Risk of Data Minorities

Recall that both the unlearned $M_{\text{unlearn}} \leftarrow \mathcal{U}(M_{\text{learn}}, D_{\text{forget}}, D_{\text{train}})$ and retrained $M_{\text{retrain}} \leftarrow \mathcal{A}(M_0, D_{\text{train}} \setminus D_{\text{forget}})$ language model depends on the choice of forget set $D_{\text{forget}}$. Whenever we estimate the privacy leakage of an unlearning method $\mathcal{U}$ via some evaluation $m$, it is important to account for potential high-risk partitions of $D_{\text{forget}}$ to ensure a comprehensive assessment of privacy risk. Unfortunately, the current LLM unlearning evaluation pipeline overlooks this critical aspect, where the partition leading to $D_{\text{forget}}$ is chosen uniformly at random (Jang et al., 2023; Chen & Yang, 2023; Yao et al., 2024; Maini et al., 2024; Zhang et al., 2024; Shi et al., 2024b). The reported privacy risk therein hence corresponds to the "average case," which may significantly underestimate the privacy risk of highly privacy-sensitive points that request unlearning. It is known in the privacy literature that some rare training samples (minorities) may have an outsized effect on model memorization compared to common training samples (majorities) (Feldman & Zhang, 2020; Carlini et al., 2022). Intuitively, a similar phenomenon persists for unlearning. See Figure 1 for an illustration of our experimental design.

Here we utilize the Enron dataset as a case study. This dataset comprises 535,703 authentic emails from 158 employees of the Enron Corporation and made public by the Federal Energy Regulatory Commission. It is a standard benchmark dataset for studying PII leakage, where the phone number is one form of PII that has been extensively studied (Lukas et al., 2023). The phone numbers here follow the format of the U.S. phone numbers (e.g., 123-4567890), with the first three digits as the area code, representing the location where the number holder applied for the number. Such information is considered sensitive as it leaks not only the phone number itself, but also the geographic information pertaining to the number holder.

Table 1: Top three most frequent and least frequent area codes within Enron dataset.

| Area code | Count |
|---|---|
| 713 (Houston) | 135,307 |
| 800 (Toll-free) | 11,902 |
| 212 (New York) | 10,739 |
| 484 (Allentown) | 1 |

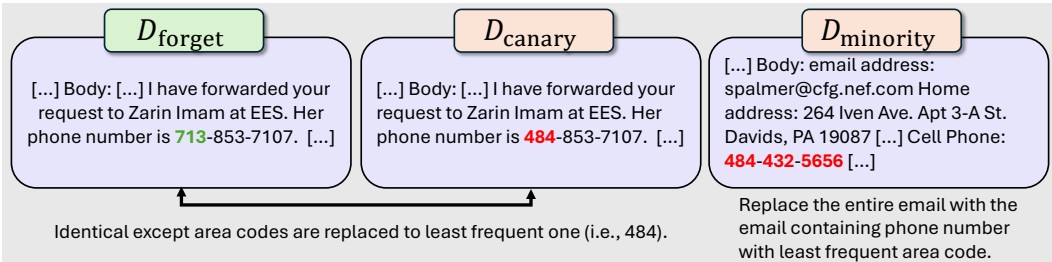

Figure 3: Examples of forget set $D_{\text{forget}}$ and how the corresponding canary set $D_{\text{canary}}$ is constructed for Enron email dataset. We also provide an example of the minority set $D_{\text{minority}}$, which consists of emails containing phone numbers with the least frequent area codes. The histogram of area codes within the Enron dataset can be found in Figure 2 and Table 1, where 713 and 484 are the most and least frequent one respectively.

Table 1 illustrates the least frequent and three most frequent area codes in the Enron dataset. The area code distribution is far from uniform. Consequently, if emails containing phone numbers are uniformly sampled for the forget set $D_{\text{forget}}$, minority data, such as emails with rare area codes like 484, are unlikely to be included due to their lower frequency. If unlearning minority data is inherently more challenging and results in greater privacy leakage, the existing evaluation pipeline may underestimate privacy risks for minorities.

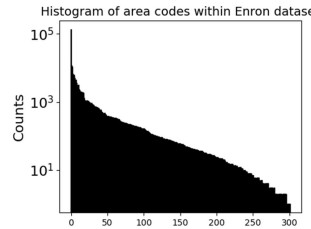

Figure 2: Area code histogram.

### 4.1 VERIFY UNDERESTIMATED PRIVACY RISKS OF MINORITY VIA CANARY INJECTION

To rigorously show that removing data from minority[3] populations indeed leads to higher unlearning privacy leakage, we design experiments based on the idea of canaries in the privacy auditing literature (Jagielski et al., 2020; Steinke et al., 2024). For simplicity, we focus on the scenario where data removal requests pertain to PIIs, where each training sample $x \in D_{\text{train}}^{(1)}$ consists of PII such as phone numbers or organization. We choose PIIs as a representative minority identifier, albeit a similar idea extends beyond PIIs. We consider the following two cases, see Figure 1 for an illustration. 1) Random: we randomly partition $D_{\text{train}}^{(1)} = D_{\text{forget}} \cup D_{\text{keep}}$ as in the standard unlearning evaluation pipeline. This leads to $M_{\text{learn}}^{(1)} \leftarrow \mathcal{A}(M_0, D_{\text{train}}^{(1)})$. 2) Canary: For the same forget set $D_{\text{forget}} = \{x_i\}_{i=1}^n$, we construct a canary set $D_{\text{canary}} = \{x_i'\}_{i=1}^n$, where each $x_i'$ is identical to $x_i$ except that the PII is replaced by the least frequent one among $D_{\text{train}}^{(1)}$. Finally, we construct a synthetic training set $D_{\text{train}}^{(2)} = D_{\text{canary}} \cup D_{\text{keep}}$, which leads to $M_{\text{learn}}^{(2)} \leftarrow \mathcal{A}(M_0, D_{\text{train}}^{(2)})$. By executing the same unlearning evaluation process for both cases $M_{\text{learn}}^{(1)}, M_{\text{learn}}^{(2)}$, we aim to show that the privacy risk for Canary is much higher than Random. By applying the same unlearning algorithm for removing $D_{\text{forget}}, D_{\text{canary}}$, we obtain the unlearned model $M_{\text{unlearn}}^{(1)}, M_{\text{unlearn}}^{(2)}$ respectively. The privacy leakage (PL) is then computed for these two cases as described in Section 3.1. The calculation of PL for Canary entails replacing $D_{\text{forget}}$ with $D_{\text{canary}}$ in Equation 1. Note that the retraining model $M_{\text{retrain}} \leftarrow \mathcal{A}(M_0, D_{\text{keep}})$ is identical for both scenarios.

An illustrative example of canary construction is provided in Figure 3. Note that for each email within $D_{\text{forget}}$ in Random, we construct the corresponding canary by only replacing its area code with the least frequent one (i.e., 484). This design is critical as we ensure the other part of the email is identical to the original email. Hence, if the privacy leakage of Canary is greater than Random, it must be due to the difference in the area code. We repeat the similar canary construction for the other PII such as email domain and year of legal judgment for different datasets.

### 4.2 QUANTIFY THE UNDERESTIMATED PRIVACY RISK OF UNLEARNING MINORITY

While our synthetic experiment on canary injection may be used to verify whether the unlearning privacy risk of minority populations is underestimated in the standard LLM unlearning evaluation pipeline, it cannot quantify the privacy risk for minorities in the real-world setting. We further design

---

[3] *Minority* here refers to any subset of the data, defined by some shared value, that is under-represented in the training set. This term is generic and may apply to any type of attribute, demographic or otherwise.

the third case aiming at quantifying the amount of underestimated privacy risk by directly choosing data to be removed containing the least frequent PII. 3) `Minority`: construct a set $D_{\text{minority}}$ that is of the same size as $D_{\text{forget}}$ in `Random`, which consists of samples with the least frequent PII within the dataset. By comparing the computed privacy risk of `Random` and `Minority`, we can quantify the amount of underestimated privacy risk for data removal from minority groups compared to the average case. If the resulting privacy risk is significantly higher than `Random`, any conclusion pertaining to unlearning efficacy drawn from `Random` can be misleading and the right to be forgotten of minorities is overlooked.

## 5 UNLEARNING METHODS

We test the following popular unlearning approaches for privacy in the literature. With a slight abuse of notation, we denote $M$ for both the model and its parameter for simplicity.

- **Random Labels (RL)** (Golatkar et al., 2020a; Yao et al., 2024): In next-token prediction, the method randomly selects from all possible token sets in disturb training on $D_{\text{forget}}$ and try to maintain performance on $D_{\text{keep}}$. The intuition for this method is that the next-token prediction of a model not seeing $D_{\text{forget}}$ should act as random guessing. However, this intuition may be inappropriate in some cases as argued in Yao et al. (2024).

- **Exact Unlearning (EUk)** and **Catastrophic Forgetting (CFk)** (Goel et al., 2022): Exact unlearning can be done by retraining the entire model from scratch on $D_{\text{keep}}$, albeit is prohibitively expansive in practice. Goel et al. (2022) propose EUk method, which retrains only the last $k$ layers of the model while freezing the other layers. As a result, it is computationally cheaper than retraining the entire model. They also propose the CFk method, which continues training the last $k$ layers on the $D_{\text{keep}}$ without retraining from scratch, while freezing the other layers.

- **Gradient Ascent (GA)** (Golatkar et al., 2020a; Graves et al., 2021; Jang et al., 2023): Gradient ascent is arguably the most popular heuristic for machine unlearning. It seeks to remove the influence of the forget set $D_{\text{forget}}$ from the trained model by reversing the gradient updates associated with $D_{\text{forget}}$. Notably, researchers have reported that gradient ascent can lead to significant model utility degradation in some cases (Ilharco et al., 2023; Pawelczyk et al., 2024a).

- **NegGrad+** (Kurmanji et al., 2024): NegGrad+ is a combination of gradient ascent on $D_{\text{forget}}$ and gradient descent on $D_{\text{keep}}$. It finetunes the current model by optimizing the following loss function:

$$\beta \cdot \hat{\mathbb{E}}_{x \sim D_{\text{keep}}}[\ell(M; x)] - (1 - \beta)\hat{\mathbb{E}}_{x \sim D_{\text{forget}}}[\ell(M; x)],$$

where $\beta \in (0, 1)$ is a hyperparameter and $\hat{\mathbb{E}}$ is the empirical expectation. The intuition is to "review" the information from $D_{\text{keep}}$ in order to prevent the model degradation due to the gradient ascent.

- **SCRUB** (Kurmanji et al., 2024): SCalable Remembering and Unlearning unBound (SCRUB) is a state-of-the-art unlearning method that leveraging a student-teacher framework. It updates the model by optimizing the objective function:

$$\hat{\mathbb{E}}_{x \sim D_{\text{keep}}}[\text{KL}(M_{\text{learn}}(x) \| M(x)) + \ell(M; x)] - \hat{\mathbb{E}}_{x \sim D_{\text{forget}}}[\text{KL}(M_{\text{learn}}(x) \| M(x))],$$

where KL is the Kullback-Leibler divergence. SCRUB shares a similar intuition with NegGrad+, which can also be viewed as a combination of gradient ascent on $D_{\text{forget}}$ and descent on $D_{\text{keep}}$. Nevertheless, instead of directly employing the original loss $\ell$, SCRUB leverages KL divergence to the original model $M_{\text{learn}}$. It provides a different regularization compared to NegGrad+.

All the above unlearning methods focus on the design of unlearning algorithm $\mathcal{U}$ and are agnostic to the learning algorithm $\mathcal{A}$. This makes them compatible with any training pipeline. On the other hand, there are unlearning solutions that design $(\mathcal{A}, \mathcal{U})$ jointly. For instance, Bourtoule et al. (2021) propose a sharding-based learning-unlearning framework SISA, which achieves exact unlearning by design. However, SISA requires training multiple models independently on partitions of the training set, which not only deviates significantly from standard machine learning pipelines but also incurs substantial memory overhead.

Chien et al. (2024a;b) recently proposed **Langevin Unlearning**, which leverages noisy gradient descent for machine unlearning. During the training process $\mathcal{A}$, it replaces the common gradient descent with DP-SGD (Abadi et al., 2016). For unlearning process $\mathcal{U}$, it finetunes the model on $D_{\text{keep}}$ with DP-SGD as well. Chien et al. (2024a) establish a smooth theoretical connection between

differential privacy and unlearning and show that Langevin Unlearning can provide a formal privacy guarantee for non-convex problems. Unfortunately, they mentioned that the resulting privacy bound is too loose to be applied in practice. We test Langevin Unlearning empirically in our experiments.

## 5.1 Enforcing the Same Computation Budget for Unlearning Methods

We categorize all methods into three groups: those that only require the forget set (RL, GA), those that only require the keep set (EUk, CFk, Langevin), and those that require both the forget and keep sets (NegGrad+, SCRUB). Since machine unlearning is about the trade-off between privacy-utility-efficiency (Guo et al., 2020; Chien et al., 2024a; Liu et al., 2024c), we carefully ensure a similar computational complexity for all tested unlearning methods when demonstrating the privacy-utility trade-off. We define a **Complexity Unit** as the gradient computation budget of one training epoch on $|D_{\text{forget}}|$ samples and limit all unlearning methods to a maximum of 10 Complexity Units. Since $|D_{\text{forget}}| = U$ is roughly 1% of $|D_{\text{train}}|$ throughout our experiments, all unlearning methods are indeed much more efficient than retraining from scratch under our setup.

For unlearning approaches leverage $D_{\text{forget}}$ only, they can unlearn for at most 10 epochs. For those leverage $D_{\text{keep}}$ only, we randomly subsample it to size $U$ for each epoch and unlearn for at most 10 epochs. For methods that leverage both $D_{\text{forget}}$ and $D_{\text{keep}}$ simultaneously, we limit their maximum unlearning epoch to 5. The situation is slightly more complicated for EUk and CFk approaches since only the last $k$ layers are trained to save computation. We randomly select $U/r$ samples from the keep set in each epoch, where $r$ is the ratio of trainable parameters in the last $k$ layers compared to the total number of parameters in the model. Our setup ensures that all tested unlearning approaches exhibit a similar unlearning computational complexity for a fair comparison. We optimize the unlearning epoch for each methods under the 10 Complexity Unit constraint by the following criterion: if the perplexity of the unlearned model on $D_{\text{train}}$ increases by more than 1 point compared to that of the initial model (No Unlearn), we stop at the first epoch where this condition is met; otherwise, we use the checkpoint from the last epoch.

## 6 Experiments

**Datasets.** We conduct our LLM unlearning evaluation pipeline on two representative PII datasets: **Enron** (Klimt & Yang, 2004) and **ECHR** (Chalkidis et al., 2019). The Enron dataset consists of corporate emails by employees that were released to the public by the Federal Energy Regulatory Commission. The ECHR dataset includes information about legal cases from the European Court of Human Rights. In our experiments, we consider specific PIIs based on their distributions within each dataset: for Enron, we consider **phone numbers** (Enron-Phone) and **email domains** (Enron-Email); for ECHR, we consider the **year of judgment** (ECHR-Year). We define data minorities based on these corresponding PIIs in each dataset. Detailed statistics for each dataset can be found in App. A.1. Note that our study focuses on instance-level unlearning, with each individual as a single record.

**General Settings.** We focus on the fine-tuning scenario, where the initial model $M_0$ is a pretrained LLM (GPT-2 (117M) (Radford et al., 2019) or Llama-2 7B (Touvron et al., 2023)). The fine-tuned model $M_{\text{learn}}$ is obtained by training $M_0$ on a dataset $D_{\text{train}}$ for 5 epochs. In the GPT-2 experiments, both the training and test sets contain 10,000 samples, subsampled from the full dataset. For Llama-2, we employ efficient fine-tuning using LoRA (Hu et al., 2021) with a rank of 16 and an alpha scaling factor of 32; both the training and test sets consist of 50,000 samples, subsampled from the entire dataset. In all cases, the forget set size is set to 100. The models are optimized using the AdamW optimizer with a constant learning rate of $10^{-5}$ and a batch size of 32, following the settings described in Shi et al. (2024b). During the unlearning process, all unlearning methods are constrained to the same computational budget—not exceeding 10 complexity units—as detailed in Section 5.1. We ensure that the unlearning complexity of each method is similar to allow for a fair comparison. Our ultimate goal in machine unlearning is to achieve a superior privacy-utility-efficiency trade-off. We utilize MIA to estimate the empirical privacy risk measured by the PL metric as described in Section 3.1. For evaluating the utility of the LLMs, we report the perplexity following standard practices in the literature (Radford et al., 2019; Zhang et al., 2022), where a lower perplexity indicates that the model is more confident in its predictions. Additional details are provided in App. B.

## 6.1 Standard Approaches Underestimate Privacy Risk for Minorities.

We report the results pertaining to the Enron-Phone, Enron-Email, and the ECHR-Year datasets. The experiment setting follows the explanation in Section 4 and further details are relegated to

Table 2: The privacy leakage (PL) for each unlearning method against different attackers for GPT-2 on the Enron-Phone dataset. The number in the parenthesis is the excess ratio of PL magnitude for cases `Canary` and `Minority` compared to `Random`, where a larger PL magnitude implies a more severe underestimation of privacy leakage in the standard unlearning evaluation (`Random`). Bold font indicates the case that the amount of underestimated privacy leakage is at least 20%.

| Method | PL (lossMIA) | | | PL (zlibMIA) | | | PL (Min-K%) | | |
|---|---|---|---|---|---|---|---|---|---|
| | Random | Canary | Minority | Random | Canary | Minority | Random | Canary | Minority |
| No Unlearn | 0.190 | **0.283 (49%↑)** | **0.340 (79%↑)** | 0.052 | **0.076 (48%↑)** | **0.064 (24%↑)** | 0.300 | **0.447 (49%↑)** | **0.524 (75%↑)** |
| RL | 0.118 | **0.191 (61%↑)** | **0.210 (77%↑)** | 0.044 | **0.067 (52%↑)** | **0.060 (37%↑)** | 0.258 | **0.401 (55%↑)** | **0.447 (73%↑)** |
| EUk | 0.027 | **0.080 (198%↑)** | **0.124 (362%↑)** | 0.035 | **0.051 (47%↑)** | **0.052 (49%↑)** | 0.092 | **0.215 (134%↑)** | **0.223 (143%↑)** |
| CFk | 0.190 | **0.278 (46%↑)** | **0.337 (77%↑)** | 0.053 | **0.075 (41%↑)** | **0.064 (21%↑)** | 0.298 | **0.435 (46%↑)** | **0.514 (73%↑)** |
| GA | 0.089 | **0.140 (57%↑)** | **0.127 (42%↑)** | 0.024 | **0.042 (73%↑)** | 0.026 (7%↑) | 0.151 | **0.242 (60%↑)** | 0.171 (13%↑) |
| NegGrad+ | 0.183 | **0.271 (48%↑)** | **0.327 (79%↑)** | 0.052 | **0.073 (42%↑)** | 0.058 (13%↑) | 0.293 | **0.435 (48%↑)** | **0.511 (74%↑)** |
| SCRUB | 0.167 | **0.251 (50%↑)** | **0.321 (92%↑)** | 0.048 | **0.070 (44%↑)** | **0.062 (28%↑)** | 0.295 | **0.450 (52%↑)** | **0.527 (78%↑)** |
| Langevin | 0.093 | **0.144 (54%↑)** | **0.157 (69%↑)** | 0.024 | **0.037 (54%↑)** | 0.027 (12%↑) | 0.160 | **0.258 (61%↑)** | **0.264 (65%↑)** |

Table 3: The privacy leakage (PL) for each unlearning method against different attackers for GPT-2 on the Enron-Email dataset.

| Method | PL (lossMIA) | | | PL (zlibMIA) | | | PL (Min-K%) | | |
|---|---|---|---|---|---|---|---|---|---|
| | Random | Canary | Minority | Random | Canary | Minority | Random | Canary | Minority |
| No Unlearn | 0.303 | **0.535 (77%↑)** | **1.145 (278%↑)** | 0.200 | **0.309 (54%↑)** | **0.262 (31%↑)** | 0.529 | **0.934 (76%↑)** | **1.468 (178%↑)** |
| RL | 0.033 | **0.153 (366%↑)** | **0.448 (1265%↑)** | 0.062 | **0.142 (127%↑)** | **0.121 (94%↑)** | 0.431 | **0.772 (79%↑)** | **1.200 (179%↑)** |
| EUk | 0.232 | **0.440 (89%↑)** | **0.582 (150%↑)** | 0.135 | **0.229 (70%↑)** | **0.152 (13%↑)** | 0.501 | **0.886 (77%↑)** | **1.034 (106%↑)** |
| CFk | 0.296 | **0.515 (74%↑)** | **1.139 (285%↑)** | 0.197 | **0.295 (49%↑)** | **0.260 (32%↑)** | 0.526 | **0.905 (72%↑)** | **1.478 (181%↑)** |
| GA | -0.279 | -0.173 (38%↓) | **0.739 (165%↑)** | -0.119 | -0.037 (69%↓) | **0.168 (41%↑)** | -0.390 | -0.304 (22%↓) | **1.034 (165%↑)** |
| NegGrad+ | 0.265 | **0.471 (77%↑)** | **1.103 (316%↑)** | 0.179 | **0.269 (50%↑)** | **0.251 (40%↑)** | 0.496 | **0.864 (74%↑)** | **1.434 (189%↑)** |
| SCRUB | 0.286 | **0.499 (74%↑)** | **1.097 (283%↑)** | 0.190 | **0.289 (52%↑)** | **0.253 (34%↑)** | 0.519 | **0.902 (74%↑)** | **1.473 (184%↑)** |
| Langevin | 0.154 | **0.319 (107%↑)** | **0.606 (293%↑)** | 0.086 | **0.178 (107%↑)** | **0.124 (44%↑)** | 0.336 | **0.645 (92%↑)** | **0.940 (180%↑)** |

Table 4: The privacy leakage (PL) for each unlearning method against different attackers for LLaMA-2 7B on the Enron-Phone dataset.

| Method | PL (lossMIA) | | | PL (zlibMIA) | | | PL (Min-K%) | | |
|---|---|---|---|---|---|---|---|---|---|
| | Random | Canary | Minority | Random | Canary | Minority | Random | Canary | Minority |
| No Unlearn | 0.060 | **0.242 (303%↑)** | **0.172 (187%↑)** | 0.034 | **0.098 (188%↑)** | **0.067 (97%↑)** | 0.076 | **0.115 (51%↑)** | **0.179 (136%↑)** |
| RL | -0.242 | -0.084 (65%↓) | -0.055 (77%↓) | -0.005 | **0.065 (1400%↑)** | **0.102 (2140%↑)** | -0.123 | -0.073 (41%↓) | **0.012 (90%↓)** |
| EUk | 0.057 | **0.246 (332%↑)** | **0.185 (225%↑)** | 0.039 | **0.106 (172%↑)** | **0.082 (110%↑)** | 0.063 | **0.132 (110%↑)** | **0.189 (200%↑)** |
| CFk | 0.057 | **0.236 (314%↑)** | **0.168 (195%↑)** | 0.032 | **0.094 (194%↑)** | **0.063 (97%↑)** | 0.072 | **0.108 (50%↑)** | **0.171 (138%↑)** |
| GA | -0.562 | -0.430 (23%↓) | -0.464 (17%↓) | -0.014 | **0.038 (371%↑)** | **0.083 (593%↑)** | -0.625 | -0.459 (27%↓) | -0.517 (17%↓) |
| NegGrad+ | -0.074 | **-0.184 (149%↑)** | -0.040 (46%↓) | -0.021 | **-0.048 (129%↑)** | -0.002 (90%↓) | -0.069 | **-0.271 (293%↑)** | -0.057 (17%↓) |
| SCRUB | 0.059 | **0.162 (175%↑)** | **0.170 (188%↑)** | 0.034 | **0.063 (85%↑)** | **0.065 (91%↑)** | 0.074 | -0.031 (58%↓) | **0.177 (139%↑)** |
| Langevin | 0.033 | **0.180 (445%↑)** | **0.104 (215%↑)** | 0.016 | **0.068 (325%↑)** | **0.036 (125%↑)** | 0.033 | **0.055 (67%↑)** | **0.091 (176%↑)** |

App. A.1. Table 2 and 4 shows that across all three attackers (lossMIA, zlibMIA, and Min-K%), all six unlearning methods and original model (no unlearning), the privacy leakage measure is significantly larger when unlearning canaries and minorities on Enron-Phone dataset for GPT-2 and Llama-2 7B respectively. Notably, in almost all cases the privacy leakage is underestimated for at least 20%. A similar phenomenon holds for the Enron-Email (Table 3, 5) and ECHR-Year (Table 7, 8 in App. C.1) datasets. It is worth noting that the amount of underestimated privacy leakage measure can be up to 68x in some cases, see Table 5. These results verify our claim that the current LLM unlearning evaluation indeed understated the privacy risk, especially for minorities. Our results call for a more careful empirical LLM unlearning evaluation, where considering canaries and minorities as we described can be an effective first step.

## 6.2 Benchmarking Unlearning Approaches under Minority-Aware Evaluation.

Motivated by our observations, we propose the minority-aware LLM unlearning evaluation. Instead of reporting the privacy leakage (PL) score under the Random case, we propose to report the **magnitude of maximum PL score** of three settings (`Random`, `Canary`, and `Minority`). This provides a

Table 5: The privacy leakage (PL) for each unlearning method against different attackers for LLaMA-2 7B on the Enron-Email dataset.

| Method | PL (lossMIA) | | | PL (zlibMIA) | | | PL (Min-K%) | | |
|---|---|---|---|---|---|---|---|---|---|
| | Random | Canary | Minority | Random | Canary | Minority | Random | Canary | Minority |
| No Unlearn | 0.050 | **0.282 (464%↑)** | **0.174 (248%↑)** | 0.046 | **0.236 (413%↑)** | **0.095 (106%↑)** | 0.064 | **0.474 (640%↑)** | **0.224 (250%↑)** |
| RL | -0.609 | -0.567 (7%↓) | -0.821 (12%↓) | -0.832 | -0.874 (5%↑) | -0.478 (43%↓) | -0.931 | -0.931 (0%) | -0.849 (9%↓) |
| EUk | 0.037 | **0.241 (551%↑)** | **0.169 (356%↑)** | 0.015 | **0.186 (1140%↑)** | **0.102 (580%↑)** | 0.040 | **0.364 (810%↑)** | **0.206 (415%↑)** |
| CFk | 0.049 | **0.264 (438%↑)** | **0.169 (245%↑)** | 0.046 | **0.220 (378%↑)** | **0.090 (96%↑)** | 0.062 | **0.436 (603%↑)** | **0.218 (251%↑)** |
| GA | -0.512 | **-0.692 (35%↑)** | 0.059 (89%↓) | -0.435 | -0.232 (47%↓) | 0.184 (58%↓) | -0.569 | -0.479 (16%↓) | 0.294 (48%↓) |
| NegGrad+ | -0.931 | -0.929 (2%↓) | -0.821 (12%↓) | -0.832 | -0.874 (5%↑) | -0.478 (43%↓) | -0.931 | -0.931 (0%) | -0.849 (9%↓) |
| SCRUB | 0.040 | **0.257 (543%↑)** | **0.174 (335%↑)** | 0.034 | **0.209 (515%↑)** | **0.095 (179%↑)** | 0.056 | **0.426 (661%↑)** | **0.224 (300%↑)** |
| Langevin | 0.022 | **0.191 (768%↑)** | **0.048 (118%↑)** | 0.020 | **0.141 (605%↑)** | 0.021 (5%↑) | 0.035 | **0.339 (868%↑)** | **0.079 (126%↑)** |

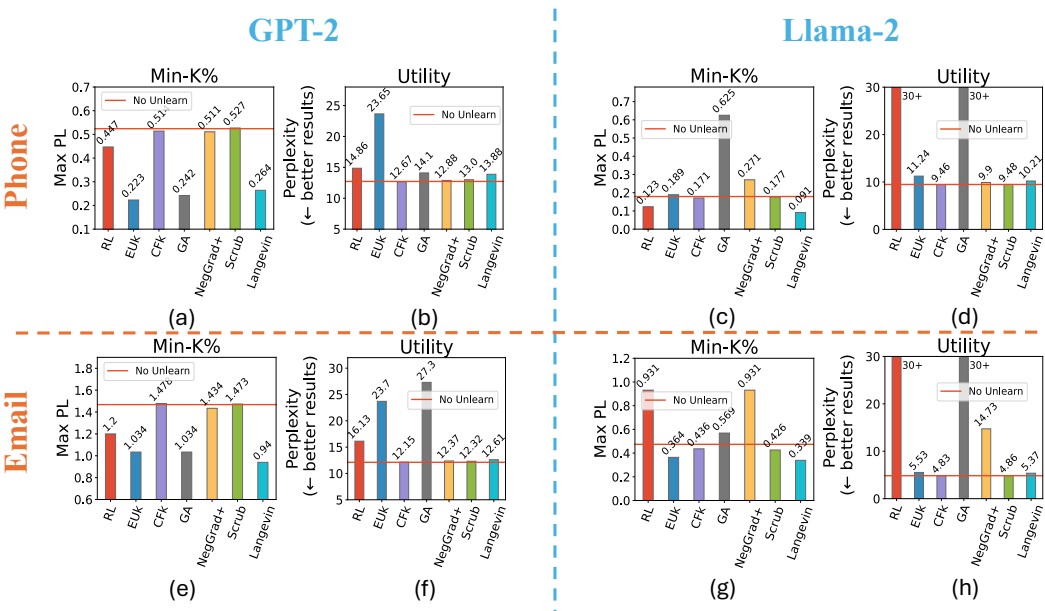

Figure 4: Benchmarking unlearning approaches via our minority-aware evaluation for GPT-2 (Left) and Llama-2 (Right) on Enron-Phone (Top) and Enron-Email (Bottom) dataset. (a),(c),(e),(g): Maximum privacy leakage (PL) over three cases (`Random`, `Canary`, and `Minority`) for Min-K% attack. (b),(d),(f),(h): Worst perplexity over the three cases of each method. More results on lossMIA and zlibMIA attackers are deferred to App. C.2.

better privacy risk estimation while keeping the entire evaluation pipeline efficient. We also report the corresponding worst-case perplexity as the utility measure for each unlearning approach. We benchmark the popular unlearning methods under our new evaluation pipeline, where the result is summarized in Fig. 4 for GPT-2 and Llama-2 on the Enron-Phone and Enron-Email datasets. See App. C.2 for additional results.

We found that Langevin Unlearning offers the best balance between privacy and utility empirically. Note that while gradient ascent has on-par performance compared to Langevin Unlearning on the Enron-Phone dataset, it significantly degrades the model utility on the Enron-Email dataset. This echoes the finding of Ilharco et al. (2023); Pawelczyk et al. (2024a) albeit for different tasks. We found that gradient ascent is inherently unstable. In contrast, unlearning methods that leverage keep set $D_{\text{keep}}$ are much more stable, including Langevin Unlearning and SCRUB.

## 6.3 ABLATION STUDIES.

In this section, we present ablation studies on the Enron-Phone dataset using the GPT-2 model, unless otherwise specified, and further results are deferred to appendix.

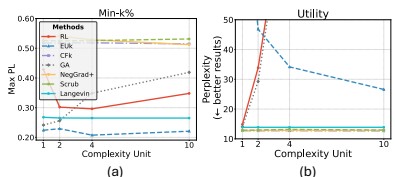
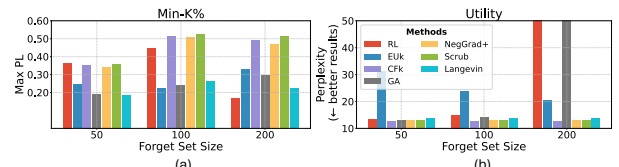

The effect of unlearning epochs for each unlearning approach. (a): Maximum PL with the attacker being Min-K%. (b): Model perplexity.

The effect of forget set size for each unlearning approach. (a): Maximum PL with the attacker being Min-K%. (Results on loss-MIA, zlibMIA are deferred to App. C.4) (b): Model perplexity.

Figure 6: Ablation studies on unlearning iterations and forget set size.

As shown in Fig.4, Langevin Unlearning and SCRUB exhibit the most stable performance, achieving relatively strong privacy-utility trade-offs. In this section, we further examine the privacy-utility trade-off curves for these two methods. For Langevin Unlearning, we vary the noise scale during training and unlearning, while in SCRUB, we adjust the weights balancing the loss and KL regularizer terms in its objective function (Sec. 5).

**Privacy-utility Trade-off.** Privacy-utility trade-off curves for both methods on Enron-Phone and Enron-Email are presented in Fig. 5. The results indicate that Langevin Unlearning (Green Line) achieves a better privacy-utility trade-off than SCRUB (Purple Line). Notably, while GA performs well on the Enron-Phone dataset, it shows a poorer trade-off on Enron-Email, underscoring its instability. Further details on the hyperparameter search for Langevin Unlearning and SCRUB are provided in App. C.5.

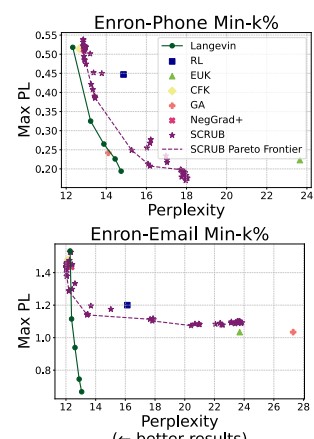

**Unlearning Iteration.** We investigate the effect of unlearning epochs on privacy (Max PL) and utility (perplexity) for each unlearning approach in Fig. 6 (Left). We observe that RL and GA are unstable in PL score. Furthermore, these two methods can lead to significant model utility degradation in terms of perplexity, where even unlearn for 2 epochs can already result in a model breakdown. This observation again demonstrates that gradient ascent, albeit being simple and popular, is not a reliable LLM unlearning solution. We should focus on stable unlearning solutions such as SCRUB and Langevin Unlearning.

Figure 5: Privacy-utility Trade-off Curves for GPT-2.

**Size of Forget Set.** In Fig. 6 (Right), we report the effect of different forget set sizes on the privacy (Max PL) and utility (Perplexity) trade-offs for each unlearning method. We find that both the GA and RL methods are highly sensitive to the forget set size, leading to significant model utility degradation and poor reliability in practice. In contrast, methods like SCRUB, Langevin Unlearning demonstrate good performance in terms of stability.

# 7 CONCLUSIONS

In this paper, we highlight a major limitation in the typical evaluation pipeline for LLM unlearning efficacy: *privacy risks of minority groups in the training data are usually underestimated*. We support this assertion with carefully crafted experiments, incorporating unlearning canaries linked to minority groups, drawing inspiration from privacy auditing research. By using personally identifiable information (PII) as a proxy for minority identifiers, we show that minority groups experience at least 20% more privacy leakage in general. Since the right to be forgotten must apply to all individuals, we call for more stringent evaluations of LLM unlearning techniques. We further benchmark existing unlearning solutions with our minority-aware unlearning evaluation for LLMs, where the popular heuristic, gradient ascent, is found to be unstable and suffers from model utility degradation in some cases. Approaches such as SCRUB and Langevin Unlearning are found to be more robust.

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

# Appendix

## CONTENTS

## A  DATASET

### A.1  DATASET DETAILS

In this paper, we examine two representative PII datasets: Enron and ECHR, described as follows:

- **Enron (Klimt & Yang, 2004).** The Enron dataset consists of 536,000 authentic emails from 158 employees of the Enron Corporation, made publicly available by the Federal Energy Regulatory Commission following an investigation. Each email typically includes the sending timestamp, sender and recipient information, a greeting, the main content, and a footer containing the sender's personal details.

- **ECHR (Chalkidis et al., 2019).** The ECHR dataset comprises case records from the European Court of Human Rights. Each record contains a series of factual lists that detail the specifics of a case. In our experiments, we further decompose these cases into individual facts, with each fact forming a distinct sample, averaging around 80 tokens in length. In total, the dataset includes around 118,000 samples.

### A.2  PII SELECTION

As outlined in Sec. 4, we selected U.S. phone numbers from the Enron dataset based on a criterion aimed at analyzing privacy risks in minority groups. To ensure the PII distribution was imbalanced, reflecting both minority and majority groups, we additionally selected two standard PII types (Lukas et al., 2023): email addresses (Enron) and years (ECHR). The distributions of these PII counts are depicted in Fig. 7.

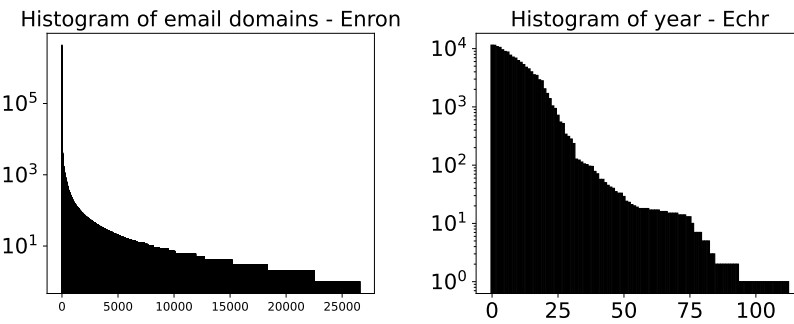

Figure 7: Histogram of email addresses and years.

### A.3  PREPROCESSING AND DATASET SPLIT

**Dataset Preprocess.** In our experiments, since the average token length of Enron samples is approximately 770, we controlled the token length of each fact to ensure the model could effectively memorize the samples. We randomly selected three coherent sentences from each sample, and if the sample contained specific PIIs of interest, we prioritized selecting sentences around them. We will keep the original samples for the ECHR dataset.

**Dataset Construction.** We begin by searching the dataset for occurrences of specific PIIs and analyzing their distribution. To form the minority set used in our `Minority` setting, we select 100 samples containing the least frequent PIIs; this set serves as our forget set. In the `Random` setting, we construct the forget set by randomly selecting 100 samples containing PIIs. To create the canary set, we replace the PIIs in the forget set (`Random` setting) with the least frequent PII found in the dataset. From the remaining data, we randomly select samples to create the training and test sets. For experiments with GPT-2 (117M), we uniformly at random selected 10,000 samples each for both the training and test sets. For Llama-2 7B, we uniformly at random selected 50,000 samples for both the training and test sets.

# B   EXPERIMENTAL DETAILS

## B.1   COMPUTE CONFIGURATIONS

All experiments were conducted using 8 NVIDIA A100 GPUs (80GB) and 14 NVIDIA RTX 6000 Ada GPUs (48GB).

## B.2   UNLEARNING EXPERIMENT SETUP

**Unlearning Algorithms.** For all unlearning methods, we use a constant learning rate of $10^{-5}$ and a batch size of 32, consistent with the fine-tuning stage. Note that some unlearning algorithms require additional hyperparameters. We follow the common designs from previous literature (Pawelczyk et al., 2024a) and detail the hyperparameter selection as follows:

- **EUK and CFK.** In our experiments, we set the number of retrained layers to $k = 3$ for both GPT-2 and Llama-2 (LoRA) models. For GPT-2, the unfrozen trainable parameters account for approximately 16% of the total parameters, while for Llama-2 7B, the unfrozen parameters account for around 10%.

- **NegGrad+.** As noted in the main text, the hyperparameter $\beta$ balances samples between $D_{\text{forget}}$ and $D_{\text{keep}}$. In these experiments, we set $\beta = 0.999$.

- **SCRUB.** In the SCRUB method, three hyperparameters are used to balance the loss function on the keep set and the KL regularizers on both the keep and forget sets. According to the definition of the objective function in Section 5, three terms are weighted sequentially by setting: $\alpha = 0.5$, $\beta = 1$, and $\gamma = 0.01$.

- **Langevin Unlearning.** The Langevin Unlearning method leverages noisy gradient descent to unlearn samples from the forget set. In our experiments, we set the Gaussian noise scale to $\sigma = 5e - 4$ (for GPT-2) and $\sigma = 5e - 3$ (for Llama-2), and the clipping norm to 1.

**Unlearning Epoch Selection.** As outlined in Section 5.1, all unlearning methods are constrained to a maximum of 10 complexity units, and the optimal epoch for each method is selected based on whether the perplexity of the unlearned model on $D_{\text{train}}$ increases by more than 1 point. Under our computational budget, methods that only require the forget set (RL, GA) are run for 10 epochs, while methods requiring both the keep and forget sets (NegGrad+, SCRUB) are limited to 5 epochs, due to the equal-sized cycling between the two sets. For methods that only require the keep set (EUK, CFK, Langevin), we use 10 epochs, with varying sample sizes for EUK and CFK, as some model parameters remain frozen. The selected epoch for each method in each experiment is detailed in Table 6.

Table 6: Epochs comparison between unlearning methods on GPT2 and LLaMA2 models.

| Unlearning Methods | GPT-2 | | Llama-2 7B | |
| --- | --- | --- | --- | --- |
| | **Enron** | **ECHR** | **Enron** | **ECHR** |
| **RL** | Epoch 1 | Epoch 1 | Epoch 1 | Epoch 1 |
| **EUk** | Epoch 10 | Epoch 10 | Epoch 10 | Epoch 10 |
| **CFk** | Epoch 10 | Epoch 10 | Epoch 10 | Epoch 10 |
| **GA** | Epoch 1 | Epoch 1 | Epoch 1 | Epoch 1 |
| **NegGrad+** | Epoch 5 | Epoch 5 | Epoch 1 | Epoch 5 |
| **SCRUB** | Epoch 5 | Epoch 5 | Epoch 5 | Epoch 5 |
| **Langevin** | Epoch 10 | Epoch 10 | Epoch 10 | Epoch 10 |

**Attack Method Hyperparameters.** We employed three attack methods in our evaluation pipeline. For lossMIA and zlibMIA, there are no hyperparameters to tune. The Min-$K\%$ method is based on the observation that non-member examples tend to have more tokens with lower likelihoods compared to member examples. In this method, the hyperparameter $K$ controls the selection of the bottom $K\%$ of tokens in each sample based on their likelihoods. Following previous recommendations in Duan et al. (2024); Shi et al. (2024a), we set $K = 20$ in our experiments.

Table 7: The privacy leakage (PL) for each unlearning method against different attackers for GPT-2 on ECHR-year datasets.

| Method | PL (lossMIA) | | | PL (zlibMIA) | | | PL (Min-K%) | | |
|---|---|---|---|---|---|---|---|---|---|
| | Random | Canary | Minority | Random | Canary | Minority | Random | Canary | Minority |
| No Unlearn | 0.198 | **0.247 (25%↑)** | **0.263 (33%↑)** | 0.086 | **0.103 (20%↑)** | **0.122 (42%↑)** | 0.213 | **0.276 (30%↑)** | **0.299 (40%↑)** |
| RL | 0.161 | **0.213 (32%↑)** | **0.234 (45%↑)** | 0.067 | **0.088 (31%↑)** | **0.086 (28%↑)** | 0.190 | **0.259 (36%↑)** | **0.257 (35%↑)** |
| EUk | 0.125 | **0.176 (41%↑)** | 0.138 (10%↑) | 0.067 | **0.088 (31%↑)** | 0.070 (4%↑) | 0.114 | **0.187 (64%↑)** | 0.135 (18%↑) |
| CFk | 0.188 | **0.234 (24%↑)** | **0.260 (38%↑)** | 0.084 | 0.095 (13%↑) | **0.120 (43%↑)** | 0.209 | **0.264 (26%↑)** | **0.295 (41%↑)** |
| GA | 0.067 | 0.027 (60%↓) | **0.105 (57%↑)** | 0.024 | 0.019 (21%↓) | **0.038 (58%↑)** | 0.090 | -0.019 (79%↓) | **0.143 (59%↑)** |
| NegGrad+ | 0.183 | **0.221 (21%↑)** | **0.247 (35%↑)** | 0.071 | **0.088 (24%↑)** | **0.112 (58%↑)** | 0.191 | **0.237 (24%↑)** | **0.274 (43%↑)** |
| SCRUB | 0.179 | **0.223 (25%↑)** | **0.253 (41%↑)** | 0.080 | **0.099 (24%↑)** | **0.116 (45%↑)** | 0.197 | **0.253 (38%↑)** | **0.289 (47%↑)** |

**Random Seed Selection.** In all our experiments, we followed the common practice and fixed our random seed to be 42.

## C  ADDITIONAL EXPERIMENTAL RESULTS

In this section, we present supplementary experimental results to further substantiate our claims in the main text.

### C.1  EXPERIMENTS ON ECHR-YEAR DATASET

In Tables 7 and 8, we report the PL scores for all three attackers across the three scenarios on the Enron-email dataset for GPT-2 and Llama-2, respectively. The results support our claim that the current LLM unlearning evaluation (`Random` setting) significantly underestimates privacy risk.

Table 8: The privacy leakage (PL) for each unlearning method against different attackers for LLaMA-2 7B on the ECHR-year dataset.

| Method | PL (lossMIA) | | | PL (zlibMIA) | | | PL (Min-K%) | | |
|---|---|---|---|---|---|---|---|---|---|
| | Random | Canary | Minority | Random | Canary | Minority | Random | Canary | Minority |
| No Unlearn | 0.056 | **0.094 (68%↑)** | **0.076 (35%↑)** | 0.030 | **0.048 (60%↑)** | **0.096 (220%↑)** | 0.067 | **0.114 (70%↑)** | **0.138 (106%↑)** |
| RL | -0.069 | 0.044 (36%↓) | **-0.532 (671%↑)** | -0.024 | **0.029 (21%↑)** | **-0.192 (700%↑)** | -0.034 | **0.070 (106%↑)** | **-0.458 (1247%↑)** |
| EUk | 0.059 | **0.084 (42%↑)** | **0.079 (34%↑)** | 0.030 | **0.044 (47%↑)** | **0.079 (163%↑)** | 0.065 | **0.110 (69%↑)** | **0.153 (135%↑)** |
| CFk | 0.056 | **0.088 (57%↑)** | **0.073 (30%↑)** | 0.028 | **0.044 (57%↑)** | **0.088 (214%↑)** | 0.063 | **0.106 (68%↑)** | **0.131 (108%↑)** |
| GA | -0.046 | **-0.376 (717%↑)** | **-0.624 (1257%↑)** | -0.016 | **-0.120 (650%↑)** | **-0.267 (1569%↑)** | -0.063 | **-0.404 (541%↑)** | **-0.574 (811%↑)** |
| NegGrad+ | 0.024 | **-0.272 (1033%↑)** | **-0.624 (2500%↑)** | 0.012 | **-0.099 (725%↑)** | **-0.235 (1858%↑)** | 0.026 | **-0.404 (1454%↑)** | **-0.663 (2449%↑)** |
| SCRUB | 0.056 | **0.094 (68%↑)** | **0.073 (30%↑)** | 0.030 | **0.048 (60%↑)** | **0.090 (200%↑)** | 0.067 | **0.112 (67%↑)** | **0.139 (107%↑)** |
| Langevin | 0.026 | **0.052 (100%↑)** | **0.041 (58%↑)** | 0.010 | **0.025 (150%↑)** | **0.046 (360%↑)** | 0.028 | **0.062 (121%↑)** | **0.078 (179%↑)** |

### C.2  MORE RESULTS ON MINORITY-AWARE EVALUATION

In this section, we present further benchmarking results for unlearning approaches under minority-aware LLM evaluation. Following the same setup as Section 6.2, Fig. 8 reports the maximum PL score under lossMIA and zlibMIA attackers on Enron-Phone and Enron-Email and Fig. 9 reports the maximum PL score and worst-case perplexity for various unlearning methods on ECHR-Year (GPT-2) dataset.

We observe that both GA and Langevin Unlearning methods maintain a favorable balance between privacy and utility. However, GA can be sensitive to the forget set size and the number of unlearning iterations (Section 6.3). In practice, the GA method should be applied with caution, whereas more stable approaches like Langevin Unlearning offer a better trade-off in terms of privacy, utility, and stability.

### C.3  FURTHER DETAILS AND RESULTS ON LANGEVIN UNLEARNING

In this section, we provide additional details and results on the Langevin Unlearning methods. As mentioned in Section 5, Langevin leverages noisy gradient descent and involves training the model on

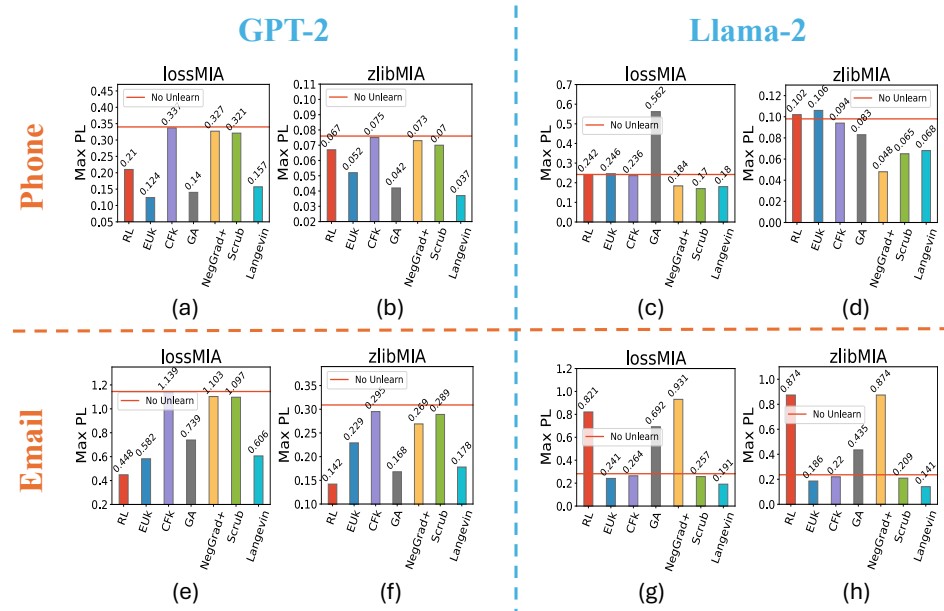

Figure 8: Benchmarking unlearning approaches via our minority-aware evaluation for GPT-2 and Llama-2 on Enron-Phone and Enron-Email Year dataset. (a),(c),(e),(g): Maximum privacy leakage (PL) over three cases (Random, Canary, and Minority) for lossMIA attack. (b),(d),(f),(h): Maximum privacy leakage (PL) over three cases (Random, Canary, and Minority) for zlibMIA attack.

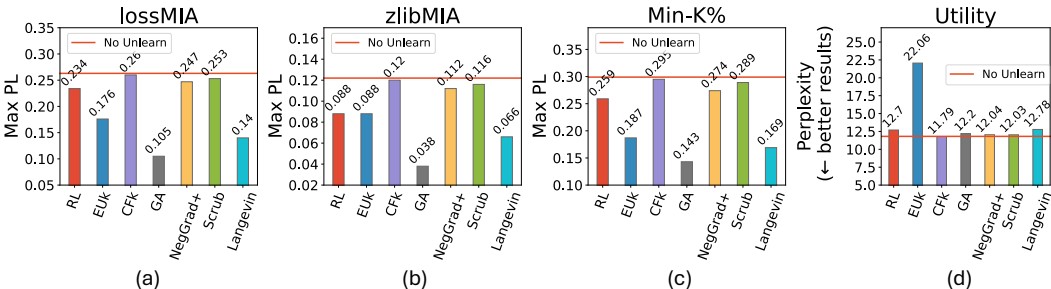

Figure 9: Benchmarking unlearning approaches via our minority-aware evaluation for GPT-2 on ECHR-year dataset. (a)-(c): Maximum privacy leakage (PL) over three cases (Random, Canary, and Minority) for lossMIA, zlibMIA, and Min-K% attacks respectively. (d): Worst perplexity over the three cases of each method.

the dataset $D_{\text{train}}$ using DP-SGD. Furthermore, Langevin conducts machine unlearning by fine-tuning the model on the dataset $D_{\text{keep}}$ with DP-SGD as well.

It is important to note that for the Langevin Unlearning method, the training process incorporates noise. Consequently, our retrain baseline is adjusted to train the initial model on $D_{\text{keep}}$ using DP-SGD for 5 epochs. Furthermore, in Table 9 and 10, we report the effectiveness of Langevin Unlearning by evaluating it against three MIA methods (lossMIA, zlibMIA, and Min-K%) across different datasets on GPT-2. These evaluations are conducted across three scenarios (Random, Canary, Minority), assessing the PL scores, the maximum PL scores and the worst-case perplexity. By comparing the results of the Noisy No Unlearn baseline (which fine-tunes the initial model with DP-SGD for 5 epochs) with those of the Langevin Unlearning method, we observe that minority scenarios (Canary, Minority) lead to significantly higher privacy leakage, and Langevin Unlearning achieves superior privacy-utility trade-offs. Additionally, in practical applications, the number of steps employing noisy gradient descent can be tailored based on the acceptable computational costs, thereby enabling potentially better privacy-utility trade-offs. This flexibility allows practitioners to

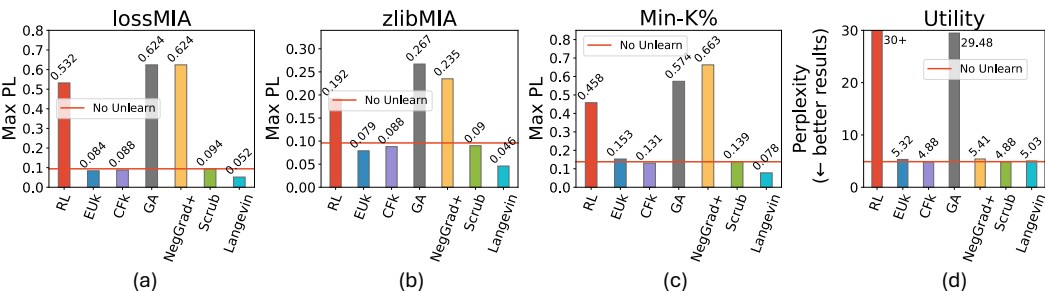

Figure 10: Benchmarking unlearning approaches via our minority-aware evaluation for Llama-2 on ECHR Year dataset. (a)-(c): Maximum privacy leakage (PL) over three cases (Random, Canary, and Minority) for lossMIA, zlibMIA, and Min-K% attacks respectively. (d): Worst perplexity over the three cases of each method.

balance the trade-off between enhanced privacy and computational efficiency according to specific application requirements.

Table 9: The privacy leakage (PL) for Langevin Unlearning against different attackers for GPT-2 on All datasets.

| Method | PL (lossMIA) | | | PL (zlibMIA) | | | PL (Min-K%) | | |
|---|---|---|---|---|---|---|---|---|---|
| | Random | Canary | Minority | Random | Canary | Minority | Random | Canary | Minority |
| | | | | Enron-phone | | | | | |
| Noisy No Unlearn | 0.097 | **0.152 (57%↑)** | **0.170 (75%↑)** | 0.024 | **0.039 (63%↑)** | **0.033 (38%↑)** | 0.164 | **0.259 (58%↑)** | **0.268 (63%↑)** |
| Langevin | 0.092 | **0.144 (57%↑)** | **0.157 (71%↑)** | 0.024 | **0.037 (54%↑)** | 0.027 (13%↑) | 0.159 | **0.259 (63%↑)** | **0.264 (66%↑)** |
| | | | | Enron-email | | | | | |
| Noisy No Unlearn | 0.156 | **0.342 (119%↑)** | **0.642 (312%↑)** | 0.102 | **0.193 (89%↑)** | **0.130 (27%↑)** | 0.344 | **0.691 (101%↑)** | **0.945 (175%↑)** |
| Langevin | 0.154 | **0.319 (107%↑)** | **0.606 (294%↑)** | 0.097 | **0.178 (84%↑)** | **0.124 (28%↑)** | 0.336 | **0.645 (92%↑)** | **0.939 (179%↑)** |
| | | | | ECHR-year | | | | | |
| Noisy No Unlearn | 0.101 | **0.152 (51%↑)** | **0.122 (21%↑)** | 0.049 | **0.067 (37%↑)** | **0.064 (31%↑)** | 0.122 | **0.180 (48%↑)** | 0.145 (19%↑) |
| Langevin | 0.103 | **0.140 (36%↑)** | **0.125 (21%↑)** | 0.049 | **0.061 (24%↑)** | **0.065 (33%↑)** | 0.117 | **0.168 (44%↑)** | **0.146 (25%↑)** |

Table 10: Maximum PL Scores and Worst-case Perplexity for Noisy No Unlearn and Langevin across Datasets on GPT-2

| Dataset | Methods | lossMIA | zlibMIA | Min-K% | Perplexity |
|---|---|---|---|---|---|
| Enron phone | Noisy No Unlearn | 0.170 | 0.039 | 0.268 | 13.87 |
| | Langevin | 0.157 (7.65%↓) | 0.037 (5.13%↓) | 0.264 (1.49%↓) | 13.88 |
| Enron email | Noisy No Unlearn | 0.642 | 0.193 | 0.945 | 12.52 |
| | Langevin | 0.606 (5.61%↓) | 0.178 (7.77%↓) | 0.939 (0.63%↓) | 12.61 |
| ECHR year | Noisy No Unlearn | 0.152 | 0.067 | 0.180 | 12.75 |
| | Langevin | 0.140 (7.89%↓) | 0.061 (8.96%↓) | 0.168 (6.67%↓) | 12.78 |

### C.4 MORE RESULTS ON FORGET SET SIZE

In this section, we report additional results on the impact of forget set size for each unlearning method, using LossMIA and ZlibMIA attackers. As shown in Fig. 11, similar to the results in the main text, both RL and GA methods are sensitive to the forget set size, whereas methods like SCRUB and Langevin Unlearning demonstrate greater stability.

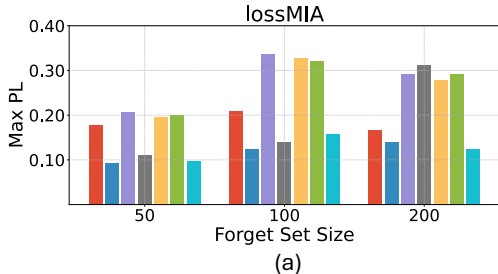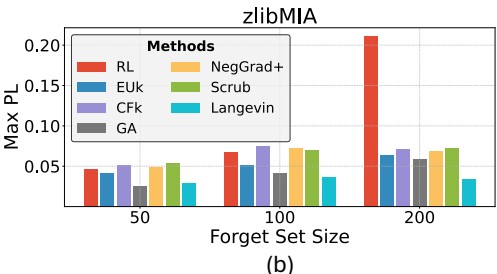

Figure 11: The effect of forget set size for each unlearning approach. (a)(b): Maximum PL over three cases (Random, Canary, Minority) with the attacker being lossMIA and zlibMIA respectively.

### C.5 MORE DETAILS ON THE PRIVACY-UTILITY TRADE-OFF CURVES FOR LANGEVIN UNLEARNING AND SCRUB METHODS

This section provides an comprehensive experiments of the privacy-utility trade-off curves for the Langevin Unlearning and SCRUB methods, as introduced in Sec. 6.3. We detailed the hyperparameters used in the ablation study as follows:

**Langevin Unlearning.** For the Langevin Unlearning method, we fix the clipping norm to 1 and vary the noise scale added during training to control the privacy-utility trade-off. In experiments with the GPT-2 model, the noise scale $\sigma$ is adjusted across the values $\{1e-4, 3e-4, 5e-4, 8e-4, 1e-3\}$. Table 17 and 18 present the AUC scores under various attackers (lossMIA, zlibMIA, Min-k%) and utility (perplexity) on the Enron-Phone and Enron-Email datasets, respectively.

**SCRUB.** The SCRUB training objective comprises the original loss $\ell$ on the keep set, along with two KL divergence regularizers on the keep and forget sets. These terms are balanced by three hyperparameters:

$$\hat{\mathbb{E}}_{x \sim D_{\text{keep}}}[\alpha \text{KL}(M_{\text{learn}}(x) \| M(x)) + \beta \ell(M; x)] - \hat{\mathbb{E}}_{x \sim D_{\text{forget}}}[\gamma \text{KL}(M_{\text{learn}}(x) \| M(x))]. \quad (2)$$

We conducted an extensive hyperparameter search, setting $\beta$ to 1 and $1e-3$ in separate configurations. For each fixed $\beta$, $\alpha$ and $\gamma$ are independently varied from $\{1e-4, 5e-4, 1e-3, 5e-3, 1e-2, 5e-2, 1e-1, 5e-1, 1\}$. Fig. 12 and 13 illustrates the resulting transition curves, showing the maximum privacy leakage (PL) for three scenarios (Random, Canary, Minority) across different attackers (lossMIA, zlibMIA, Min-k%) and utility (perplexity) metrics on both Enron-Phone and Enron-Email datasets. The transition curves highlight how SCRUB's performance depends on balancing the three objective terms. Notably, when the KL regularizer weight on the keep set is greater than or equal to that on the forget set, SCRUB achieves relatively high utility, albeit with increased privacy leakage. Besides, we observe in our experiments that the zlibMIA attacker fails to capture the inherent privacy-utility trade-off for SCRUB as demonstrated in the transition curves.

We further report the privacy-utility trade-off curves for both methods under attacker being lossMIA and zlibMIA. Similar to the results demonstrated in Sec. 6.3, Langevin Unlearning method achieves the best trade-off performance over SCRUB method.

### C.6 RESULTS ON AUC SCORES, PERPLEXITY ACROSS DIFFERENT MODELS AND DATASETS.

We further report the AUC scores under different attackers (lossMIA, zlibMIA, Min-K%) and utility (perplexity) over holdout test set $D_{\text{test}}$ for GPT-2 and Llama-2 in Table. 11-16.

**Discussion on TPR@low FPR Metric:** Note that aside from the AUC score, a commonly reported metric for privacy evaluation is **TPR@low FPR** (Carlini et al., 2022), where the low FPR is often set to 0.01. However, in our scenario, the canary size is set to 100 (1% of the total training set size). At FPR = 0.01, the TPR would be calculated based on only a few canary samples, making the overall score very coarse. To avoid the impact of this coarse granularity on our experimental results, we primarily focus on the AUC score.

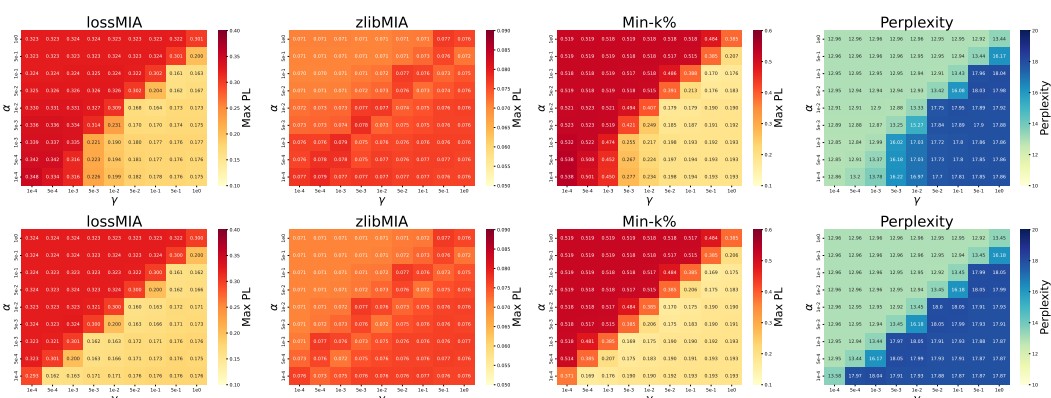

Figure 12: Privacy-utility transition curves for Enron-Phone dataset with hyperparameter $\beta = 1$ (Top) and hyperparameter $\beta = 1e - 3$ (Bottom).

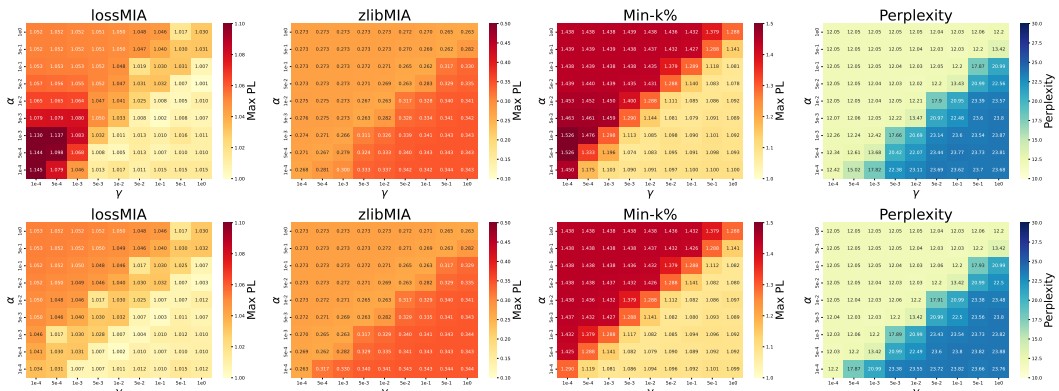

Figure 13: Privacy-utility transition curves for Enron-Email dataset with hyperparameter $\beta = 1$ (Top) and hyperparameter $\beta = 1e - 3$ (Bottom).

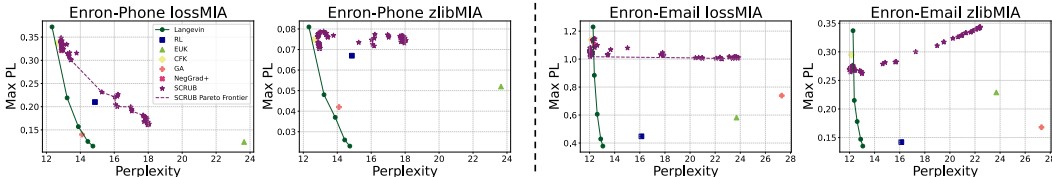

Figure 14: Privacy-utility trade-off curves under lossMIA and zlibMIA.

Table 11: AUC Scores for MIA and Perplexity across Three Settings for GPT-2 on Enron (Phone Numbers)

| | AUC - LossMIA | | | AUC - ZlibMIA | | |
|---|---|---|---|---|---|---|
| | **Random** | **Canary** | **Minority** | **Random** | **Canary** | **Minority** |
| **No Unlearn** | 0.533 | 0.531 | 0.422 | 0.694 | 0.693 | 0.619 |
| **Retrain** | 0.448 | 0.414 | 0.315 | 0.660 | 0.644 | 0.582 |
| **Noisy No Unlearn** | 0.474 | 0.464 | 0.365 | 0.678 | 0.674 | 0.606 |
| **Noisy Retrain** | 0.432 | 0.403 | 0.312 | 0.662 | 0.649 | 0.587 |
| **Unlearning Methods** | | | | | | |
| **Random Label** | 0.501 | 0.493 | 0.381 | 0.689 | 0.687 | 0.617 |
| **Langevin** | 0.472 | 0.461 | 0.361 | 0.678 | 0.673 | 0.603 |
| **EUk** | 0.460 | 0.447 | 0.354 | 0.683 | 0.677 | 0.612 |
| **CFk** | 0.533 | 0.529 | 0.421 | 0.695 | 0.692 | 0.619 |
| **Gradient Ascent** | 0.488 | 0.472 | 0.355 | 0.676 | 0.671 | 0.597 |
| **NegGrad+** | 0.530 | 0.526 | 0.418 | 0.694 | 0.691 | 0.616 |
| **SCRUB** | 0.523 | 0.518 | 0.416 | 0.692 | 0.689 | 0.618 |
| | AUC - Min-K% | | | Perplexity | | |
| | **Random** | **Canary** | **Minority** | **Random** | **Canary** | **Minority** |
| **No Unlearn** | 0.594 | 0.592 | 0.471 | 12.72 | 12.72 | 12.72 |
| **Retrain** | 0.457 | 0.409 | 0.309 | 12.74 | 12.74 | 12.74 |
| **Noisy No Unlearn** | 0.518 | 0.511 | 0.393 | 13.84 | 13.85 | 13.87 |
| **Noisy Retrain** | 0.445 | 0.406 | 0.310 | 13.84 | 13.84 | 13.83 |
| **Unlearning Methods** | | | | | | |
| **Random Label** | 0.575 | 0.573 | 0.447 | 14.49 | 14.41 | 14.86 |
| **Langevin** | 0.516 | 0.511 | 0.392 | 13.84 | 13.88 | 13.88 |
| **EUk** | 0.499 | 0.497 | 0.378 | 23.64 | 23.65 | 23.60 |
| **CFk** | 0.593 | 0.587 | 0.468 | 12.67 | 12.67 | 12.67 |
| **Gradient Ascent** | 0.526 | 0.508 | 0.362 | 13.22 | 13.20 | 14.10 |
| **NegGrad+** | 0.591 | 0.587 | 0.467 | 12.86 | 12.86 | 12.88 |
| **SCRUB** | 0.592 | 0.593 | 0.472 | 13.00 | 12.98 | 12.96 |

Table 12: AUC Scores for MIA and Perplexity across Three Settings for GPT-2 on Enron (Email)

| | AUC - LossMIA | | | AUC - ZlibMIA | | |
|---|---|---|---|---|---|---|
| | **Random** | **Canary** | **Minority** | **Random** | **Canary** | **Minority** |
| **No Unlearn** | 0.555 | 0.551 | 0.354 | 0.462 | 0.462 | 0.563 |
| **Retrain** | 0.426 | 0.359 | 0.165 | 0.385 | 0.353 | 0.446 |
| **Noisy No Unlearn** | 0.488 | 0.483 | 0.271 | 0.422 | 0.421 | 0.512 |
| **Noisy Retrain** | 0.422 | 0.360 | 0.165 | 0.383 | 0.353 | 0.453 |
| | **Unlearning Methods** | | | | | |
| **Random Label** | 0.440 | 0.414 | 0.239 | 0.409 | 0.403 | 0.500 |
| **Langevin** | 0.487 | 0.475 | 0.265 | 0.420 | 0.416 | 0.509 |
| **EUk** | 0.525 | 0.517 | 0.261 | 0.437 | 0.434 | 0.514 |
| **CFk** | 0.552 | 0.544 | 0.353 | 0.461 | 0.457 | 0.562 |
| **Gradient Ascent** | 0.307 | 0.297 | 0.287 | 0.339 | 0.340 | 0.521 |
| **NegGrad+** | 0.539 | 0.528 | 0.347 | 0.454 | 0.448 | 0.558 |
| **SCRUB** | 0.548 | 0.538 | 0.346 | 0.458 | 0.455 | 0.559 |
| | AUC - Min-K% | | | Perplexity | | |
| | **Random** | **Canary** | **Minority** | **Random** | **Canary** | **Minority** |
| **No Unlearn** | 0.607 | 0.611 | 0.506 | 12.11 | 12.11 | 12.12 |
| **Retrain** | 0.397 | 0.316 | 0.205 | 12.19 | 12.19 | 12.19 |
| **Noisy No Unlearn** | 0.516 | 0.519 | 0.387 | 12.51 | 12.52 | 12.50 |
| **Noisy Retrain** | 0.384 | 0.307 | 0.199 | 12.48 | 12.48 | 12.48 |
| | **Unlearning Methods** | | | | | |
| **Random Label** | 0.568 | 0.560 | 0.451 | 15.87 | 16.13 | 13.54 |
| **Langevin** | 0.513 | 0.505 | 0.386 | 12.58 | 12.58 | 12.61 |
| **EUk** | 0.596 | 0.596 | 0.417 | 23.58 | 23.70 | 23.58 |
| **CFk** | 0.606 | 0.602 | 0.508 | 12.14 | 12.15 | 12.15 |
| **Gradient Ascent** | 0.242 | 0.220 | 0.417 | 27.30 | 21.86 | 14.53 |
| **NegGrad+** | 0.594 | 0.589 | 0.499 | 12.34 | 12.34 | 12.37 |
| **SCRUB** | 0.603 | 0.601 | 0.507 | 12.15 | 12.15 | 12.32 |

Table 13: AUC Scores for MIA and Perplexity across Three Settings for GPT-2 on ECHR (Year)

| | AUC - LossMIA | | | AUC - ZlibMIA | | |
|---|---|---|---|---|---|---|
| | Random | Canary | Minority | Random | Canary | Minority |
| **No Unlearn** | 0.661 | 0.650 | 0.658 | 0.532 | 0.524 | 0.560 |
| **Retrain** | 0.552 | 0.521 | 0.521 | 0.490 | 0.475 | 0.499 |
| **Noisy No Unlearn** | 0.600 | 0.592 | 0.581 | 0.514 | 0.509 | 0.535 |
| **Noisy Retrain** | 0.545 | 0.514 | 0.518 | 0.490 | 0.477 | 0.503 |
| **Unlearning Methods** | | | | | | |
| **Random Label** | 0.641 | 0.632 | 0.643 | 0.523 | 0.517 | 0.542 |
| **Langevin** | 0.601 | 0.586 | 0.583 | 0.514 | 0.506 | 0.536 |
| **EUk** | 0.621 | 0.613 | 0.593 | 0.523 | 0.517 | 0.534 |
| **CFk** | 0.656 | 0.643 | 0.656 | 0.531 | 0.520 | 0.559 |
| **Gradient Ascent** | 0.589 | 0.535 | 0.576 | 0.502 | 0.484 | 0.518 |
| **NegGrad+** | 0.653 | 0.636 | 0.650 | 0.525 | 0.517 | 0.555 |
| **SCRUB** | 0.651 | 0.637 | 0.653 | 0.529 | 0.522 | 0.557 |
| | AUC - Min-K% | | | Perplexity | | |
| | Random | Canary | Minority | Random | Canary | Minority |
| **No Unlearn** | 0.671 | 0.661 | 0.673 | 11.81 | 11.81 | 11.81 |
| **Retrain** | 0.553 | 0.518 | 0.518 | 11.82 | 11.82 | 11.82 |
| **Noisy No Unlearn** | 0.615 | 0.609 | 0.586 | 12.74 | 12.74 | 12.75 |
| **Noisy Retrain** | 0.548 | 0.516 | 0.512 | 12.73 | 12.73 | 12.73 |
| **Unlearning Methods** | | | | | | |
| **Random Label** | 0.658 | 0.652 | 0.651 | 12.70 | 12.70 | 12.67 |
| **Langevin** | 0.612 | 0.603 | 0.587 | 12.78 | 12.78 | 12.78 |
| **EUk** | 0.616 | 0.615 | 0.588 | 22.04 | 22.06 | 22.00 |
| **CFk** | 0.669 | 0.655 | 0.671 | 11.78 | 11.78 | 11.79 |
| **Gradient Ascent** | 0.603 | 0.508 | 0.592 | 12.11 | 12.20 | 12.11 |
| **NegGrad+** | 0.659 | 0.641 | 0.660 | 12.01 | 12.04 | 12.03 |
| **SCRUB** | 0.662 | 0.649 | 0.668 | 12.02 | 12.03 | 12.01 |

Table 14: AUC Scores for MIA and Perplexity across Three Settings for Llama-2 on Enron (Phone Number)

| | AUC - LossMIA | | | AUC - ZlibMIA | | |
|---|---|---|---|---|---|---|
| | **Random** | **Canary** | **Minority** | **Random** | **Canary** | **Minority** |
| **No Unlearn** | 0.614 | 0.606 | 0.551 | 0.578 | 0.571 | 0.575 |
| **Retrain** | 0.579 | 0.488 | 0.470 | 0.559 | 0.520 | 0.539 |
| **Noisy No Unlearn** | 0.605 | 0.598 | 0.539 | 0.578 | 0.572 | 0.576 |
| **Noisy Retrain** | 0.584 | 0.500 | 0.482 | 0.568 | 0.533 | 0.554 |
| **Unlearning Methods** | | | | | | |
| **Random Label** | 0.439 | 0.447 | 0.444 | 0.556 | 0.554 | 0.594 |
| **Langevin** | 0.603 | 0.590 | 0.532 | 0.577 | 0.569 | 0.574 |
| **EUk** | 0.612 | 0.608 | 0.557 | 0.581 | 0.575 | 0.583 |
| **CFk** | 0.612 | 0.603 | 0.549 | 0.577 | 0.569 | 0.573 |
| **Gradient Ascent** | 0.253 | 0.278 | 0.252 | 0.551 | 0.540 | 0.584 |
| **NegGrad+** | 0.536 | 0.398 | 0.451 | 0.547 | 0.495 | 0.538 |
| **SCRUB** | 0.613 | 0.567 | 0.550 | 0.578 | 0.553 | 0.574 |
| | AUC - Min-K% | | | Perplexity | | |
| | **Random** | **Canary** | **Minority** | **Random** | **Canary** | **Minority** |
| **No Unlearn** | 0.611 | 0.610 | 0.579 | 9.45 | 9.47 | 9.48 |
| **Retrain** | 0.568 | 0.547 | 0.491 | 9.48 | 9.48 | 9.48 |
| **Noisy No Unlearn** | 0.600 | 0.602 | 0.570 | 10.21 | 10.21 | 10.21 |
| **Noisy Retrain** | 0.579 | 0.565 | 0.516 | 10.19 | 10.19 | 10.19 |
| **Unlearning Methods** | | | | | | |
| **Random Label** | 0.498 | 0.507 | 0.497 | 2124 | 2559 | 2445 |
| **Langevin** | 0.598 | 0.596 | 0.563 | 10.20 | 10.21 | 10.20 |
| **EUk** | 0.604 | 0.619 | 0.584 | 11.00 | 11.24 | 10.82 |
| **CFk** | 0.609 | 0.606 | 0.575 | 9.43 | 9.45 | 9.46 |
| **Gradient Ascent** | 0.213 | 0.296 | 0.237 | 4e9 | 8e9 | 2e9 |
| **NegGrad+** | 0.529 | 0.399 | 0.463 | 9.82 | 9.90 | 9.85 |
| **SCRUB** | 0.610 | 0.530 | 0.578 | 9.45 | 9.48 | 9.48 |

Table 15: AUC Scores for MIA and Perplexity across Three Settings for Llama-2 on Enron (Email)

| | AUC - LossMIA | | | AUC - ZlibMIA | | |
|---|---|---|---|---|---|---|
| | **Random** | **Canary** | **Minority** | **Random** | **Canary** | **Minority** |
| **No Unlearn** | 0.628 | 0.613 | 0.418 | 0.548 | 0.539 | 0.451 |
| **Retrain** | 0.598 | 0.478 | 0.356 | 0.524 | 0.436 | 0.412 |
| **Noisy No Unlearn** | 0.594 | 0.572 | 0.393 | 0.515 | 0.502 | 0.443 |
| **Retrain** | 0.579 | 0.470 | 0.375 | 0.503 | 0.433 | 0.434 |
| **Unlearning Methods** | | | | | | |
| **Random Label** | 0.234 | 0.207 | 0.394 | 0.333 | 0.333 | 0.458 |
| **Langevin** | 0.592 | 0.560 | 0.393 | 0.513 | 0.494 | 0.443 |
| **EUk** | 0.620 | 0.593 | 0.416 | 0.532 | 0.517 | 0.454 |
| **CFk** | 0.627 | 0.604 | 0.416 | 0.548 | 0.532 | 0.449 |
| **Gradient Ascent** | 0.292 | 0.147 | 0.377 | 0.296 | 0.335 | 0.488 |
| **NegGrad+** | 0.041 | 0.034 | 0.064 | 0.088 | 0.055 | 0.215 |
| **SCRUB** | 0.622 | 0.601 | 0.418 | 0.542 | 0.527 | 0.451 |
| | AUC - Min-K% | | | Perplexity | | |
| | **Random** | **Canary** | **Minority** | **Random** | **Canary** | **Minority** |
| **No Unlearn** | 0.632 | 0.619 | 0.421 | 4.83 | 4.84 | 4.84 |
| **Retrain** | 0.594 | 0.420 | 0.344 | 4.84 | 4.84 | 4.84 |
| **Noisy No Unlearn** | 0.592 | 0.569 | 0.397 | 5.38 | 5.38 | 5.38 |
| **Noisy Retrain** | 0.570 | 0.410 | 0.368 | 5.39 | 5.39 | 5.39 |
| **Unlearning Methods** | | | | | | |
| **Random Label** | 0.230 | 0.181 | 0.330 | 730 | 542 | 255 |
| **Langevin** | 0.590 | 0.549 | 0.397 | 5.36 | 5.36 | 5.37 |
| **EUk** | 0.618 | 0.573 | 0.415 | 5.42 | 5.53 | 5.37 |
| **CFk** | 0.631 | 0.603 | 0.419 | 4.83 | 4.83 | 4.83 |
| **Gradient Ascent** | 0.256 | 0.219 | 0.445 | 4e8 | 6e12 | 6e12 |
| **NegGrad+** | 0.041 | 0.029 | 0.052 | 12.15 | 14.73 | 6.20 |
| **SCRUB** | 0.627 | 0.599 | 0.421 | 4.86 | 4.86 | 4.86 |

Table 16: AUC Scores for MIA and Perplexity across Three Settings for Llama-2 on ECHR (Year)

| | AUC - LossMIA | | | AUC - ZlibMIA | | |
|---|---|---|---|---|---|---|
| | Random | Canary | Minority | Random | Canary | Minority |
| **No Unlearn** | 0.570 | 0.547 | 0.726 | 0.513 | 0.499 | 0.513 |
| **Retrain** | 0.540 | 0.500 | 0.675 | 0.498 | 0.476 | 0.468 |
| **Noisy No Unlearn** | 0.556 | 0.532 | 0.721 | 0.504 | 0.491 | 0.510 |
| **Noisy Retrain** | 0.541 | 0.502 | 0.685 | 0.498 | 0.476 | 0.476 |
| **Unlearning Methods** | | | | | | |
| **Random Label** | 0.503 | 0.522 | 0.316 | 0.486 | 0.490 | 0.378 |
| **Langevin** | 0.555 | 0.528 | 0.715 | 0.503 | 0.488 | 0.498 |
| **EUk** | 0.572 | 0.542 | 0.728 | 0.513 | 0.497 | 0.505 |
| **CFk** | 0.570 | 0.544 | 0.724 | 0.512 | 0.497 | 0.509 |
| **Gradient Ascent** | 0.515 | 0.312 | 0.254 | 0.490 | 0.419 | 0.343 |
| **NegGrad+** | 0.553 | 0.364 | 0.254 | 0.504 | 0.429 | 0.358 |
| **SCRUB** | 0.570 | 0.547 | 0.724 | 0.513 | 0.499 | 0.510 |
| | AUC - Min-K% | | | Perplexity | | |
| | Random | Canary | Minority | Random | Canary | Minority |
| **No Unlearn** | 0.573 | 0.559 | 0.686 | 4.89 | 4.89 | 4.89 |
| **Retrain** | 0.537 | 0.502 | 0.603 | 4.89 | 4.89 | 4.89 |
| **Noisy No Unlearn** | 0.553 | 0.541 | 0.675 | 5.03 | 5.03 | 5.02 |
| **Noisy Retrain** | 0.537 | 0.504 | 0.616 | 5.02 | 5.02 | 5.02 |
| **Unlearning Methods** | | | | | | |
| **Random Label** | 0.519 | 0.537 | 0.327 | 90 | 129 | 127.43 |
| **Langevin** | 0.552 | 0.535 | 0.664 | 5.03 | 5.03 | 5.03 |
| **EUk** | 0.572 | 0.557 | 0.695 | 5.27 | 5.21 | 5.32 |
| **CFk** | 0.571 | 0.555 | 0.682 | 4.87 | 4.88 | 4.87 |
| **Gradient Ascent** | 0.503 | 0.299 | 0.257 | 7.94 | 10.68 | 29.48 |
| **NegGrad+** | 0.551 | 0.299 | 0.203 | 4.96 | 5.10 | 5.41 |
| **SCRUB** | 0.573 | 0.558 | 0.687 | 4.88 | 4.88 | 4.87 |

Table 17: AUC Scores for MIA and Perplexity across Three Settings for GPT-2 on Enron (Phone Number) for Noisy Learning

| | AUC - LossMIA | | | AUC - ZlibMIA | | |
|---|---|---|---|---|---|---|
| | **Random** | **Canary** | **Minority** | **Random** | **Canary** | **Minority** |
| **Noisy No Unlearn 1e-4** | 0.541 | 0.536 | 0.427 | 0.699 | 0.697 | 0.621 |
| **Noisy Retrain 1e-4** | 0.446 | 0.412 | 0.309 | 0.658 | 0.643 | 0.577 |
| **Langevin 1e-4** | 0.540 | 0.535 | 0.424 | 0.698 | 0.695 | 0.619 |
| **Noisy No Unlearn 3e-4** | 0.492 | 0.484 | 0.381 | 0.682 | 0.678 | 0.607 |
| **Noisy Retrain 3e-4** | 0.436 | 0.405 | 0.311 | 0.660 | 0.647 | 0.583 |
| **Langevin 3e-4** | 0.492 | 0.483 | 0.379 | 0.682 | 0.678 | 0.606 |
| **Noisy No Unlearn 5e-4** | 0.474 | 0.464 | 0.365 | 0.678 | 0.674 | 0.606 |
| **Noisy Retrain 5e-4** | 0.432 | 0.403 | 0.312 | 0.662 | 0.649 | 0.587 |
| **Langevin 5e-4** | 0.472 | 0.461 | 0.361 | 0.678 | 0.673 | 0.603 |
| **Noisy No Unlearn 8e-4** | 0.463 | 0.450 | 0.352 | 0.677 | 0.672 | 0.606 |
| **Noisy Retrain 8e-4** | 0.428 | 0.401 | 0.311 | 0.664 | 0.653 | 0.590 |
| **Langevin 8e-4** | 0.460 | 0.448 | 0.350 | 0.676 | 0.670 | 0.604 |
| **Noisy No Unlearn 1e-3** | 0.459 | 0.446 | 0.348 | 0.677 | 0.671 | 0.606 |
| **Noisy Retrain 1e-3** | 0.427 | 0.400 | 0.311 | 0.665 | 0.655 | 0.592 |
| **Langevin 1e-3** | 0.456 | 0.443 | 0.347 | 0.675 | 0.670 | 0.604 |
| | **AUC - Min-K%** | | | **Perplexity** | | |
| | **Random** | **Canary** | **Minority** | **Random** | **Canary** | **Minority** |
| **Noisy No Unlearn 1e-4** | 0.599 | 0.599 | 0.465 | 12.26 | 12.24 | 12.27 |
| **Noisy Retrain 1e-4** | 0.454 | 0.410 | 0.303 | 12.25 | 12.25 | 12.25 |
| **Langevin 1e-4** | 0.598 | 0.596 | 0.460 | 12.32 | 12.32 | 12.33 |
| **Noisy No Unlearn 3e-4** | 0.538 | 0.535 | 0.409 | 13.20 | 13.20 | 13.21 |
| **Noisy Retrain 3e-4** | 0.447 | 0.405 | 0.308 | 13.19 | 13.19 | 13.19 |
| **Langevin 3e-4** | 0.538 | 0.535 | 0.408 | 13.24 | 13.24 | 13.22 |
| **Noisy No Unlearn 5e-4** | 0.518 | 0.511 | 0.393 | 13.84 | 13.85 | 13.87 |
| **Noisy Retrain 5e-4** | 0.445 | 0.406 | 0.310 | 13.84 | 13.84 | 13.83 |
| **Langevin 5e-4** | 0.516 | 0.511 | 0.392 | 13.84 | 13.88 | 13.88 |
| **Noisy No Unlearn 8e-4** | 0.505 | 0.498 | 0.378 | 14.40 | 14.39 | 14.39 |
| **Noisy Retrain 8e-4** | 0.442 | 0.405 | 0.313 | 14.42 | 14.42 | 14.42 |
| **Langevin 8e-4** | 0.502 | 0.497 | 0.376 | 14.44 | 14.44 | 14.43 |
| **Noisy No Unlearn 1e-3** | 0.500 | 0.492 | 0.374 | 14.71 | 14.70 | 14.70 |
| **Noisy Retrain 1e-3** | 0.440 | 0.408 | 0.313 | 14.73 | 14.73 | 14.73 |
| **Langevin 1e-3** | 0.497 | 0.487 | 0.371 | 14.74 | 14.74 | 14.73 |

Table 18: AUC Scores for MIA and Perplexity across Three Settings for GPT-2 on Enron (Phone Email) for Noisy Learning

| | AUC - LossMIA | | | AUC - ZlibMIA | | |
|---|---|---|---|---|---|---|
| | **Random** | **Canary** | **Minority** | **Random** | **Canary** | **Minority** |
| **Noisy No Unlearn 1e-4** | 0.584 | 0.587 | 0.369 | 0.480 | 0.483 | 0.578 |
| **Noisy Retrain 1e-4** | 0.431 | 0.362 | 0.165 | 0.387 | 0.356 | 0.447 |
| **Langevin 1e-4** | 0.584 | 0.578 | 0.368 | 0.479 | 0.476 | 0.579 |
| **Noisy No Unlearn 3e-4** | 0.512 | 0.511 | 0.311 | 0.436 | 0.436 | 0.535 |
| **Noisy Retrain 3e-4** | 0.424 | 0.360 | 0.164 | 0.384 | 0.353 | 0.449 |
| **Langevin 3e-4** | 0.510 | 0.499 | 0.309 | 0.432 | 0.429 | 0.534 |
| **Noisy No Unlearn 5e-4** | 0.488 | 0.483 | 0.271 | 0.422 | 0.421 | 0.512 |
| **Noisy Retrain 5e-4** | 0.422 | 0.360 | 0.165 | 0.383 | 0.353 | 0.453 |
| **Langevin 5e-4** | 0.487 | 0.475 | 0.265 | 0.420 | 0.416 | 0.509 |
| **Noisy No Unlearn 8e-4** | 0.470 | 0.464 | 0.248 | 0.414 | 0.412 | 0.503 |
| **Noisy Retrain 8e-4** | 0.417 | 0.357 | 0.170 | 0.383 | 0.354 | 0.459 |
| **Langevin 8e-4** | 0.471 | 0.457 | 0.243 | 0.411 | 0.406 | 0.500 |
| **Noisy No Unlearn 1e-3** | 0.463 | 0.456 | 0.243 | 0.410 | 0.408 | 0.500 |
| **Noisy Retrain 1e-3** | 0.414 | 0.354 | 0.172 | 0.381 | 0.355 | 0.462 |
| **Langevin 1e-3** | 0.462 | 0.450 | 0.237 | 0.408 | 0.403 | 0.499 |
| | AUC - Min-K% | | | Perplexity | | |
| | **Random** | **Canary** | **Minority** | **Random** | **Canary** | **Minority** |
| **Noisy No Unlearn 1e-4** | 0.651 | 0.661 | 0.528 | 12.18 | 12.25 | 12.16 |
| **Noisy Retrain 1e-4** | 0.407 | 0.318 | 0.210 | 12.25 | 12.25 | 12.25 |
| **Langevin 1e-4** | 0.655 | 0.655 | 0.532 | 12.27 | 12.28 | 12.27 |
| **Noisy No Unlearn 3e-4** | 0.553 | 0.562 | 0.420 | 12.31 | 12.34 | 12.28 |
| **Noisy Retrain 3e-4** | 0.390 | 0.311 | 0.199 | 12.23 | 12.23 | 12.23 |
| **Langevin 3e-4** | 0.552 | 0.547 | 0.421 | 12.38 | 12.39 | 12.38 |
| **Noisy No Unlearn 5e-4** | 0.516 | 0.519 | 0.387 | 12.51 | 12.52 | 12.50 |
| **Noisy Retrain 5e-4** | 0.384 | 0.307 | 0.199 | 12.48 | 12.48 | 12.48 |
| **Langevin 5e-4** | 0.513 | 0.505 | 0.386 | 12.58 | 12.58 | 12.61 |
| **Noisy No Unlearn 8e-4** | 0.478 | 0.483 | 0.355 | 12.83 | 12.80 | 12.82 |
| **Noisy Retrain 8e-4** | 0.369 | 0.298 | 0.200 | 12.84 | 12.84 | 12.84 |
| **Langevin 8e-4** | 0.477 | 0.469 | 0.349 | 12.87 | 12.86 | 12.89 |
| **Noisy No Unlearn 1e-3** | 0.465 | 0.467 | 0.341 | 13.01 | 12.98 | 13.01 |
| **Noisy Retrain 1e-3** | 0.365 | 0.296 | 0.201 | 13.02 | 13.02 | 13.02 |
| **Langevin 1e-3** | 0.462 | 0.453 | 0.335 | 13.04 | 13.03 | 13.06 |

# D    ADDITIONAL EXPERIMENTS FOR REBUTTAL

We present additional experiments to further address the questions raised by the reviewer.

**To Reviewer qsaJ:**

† **Experiments to W2.** Langevin Unlearning demonstrates a good privacy-utility trade-off, whereas Gradient Ascent and Random Label methods exhibit instability across different models (See Fig. 15 and 16).

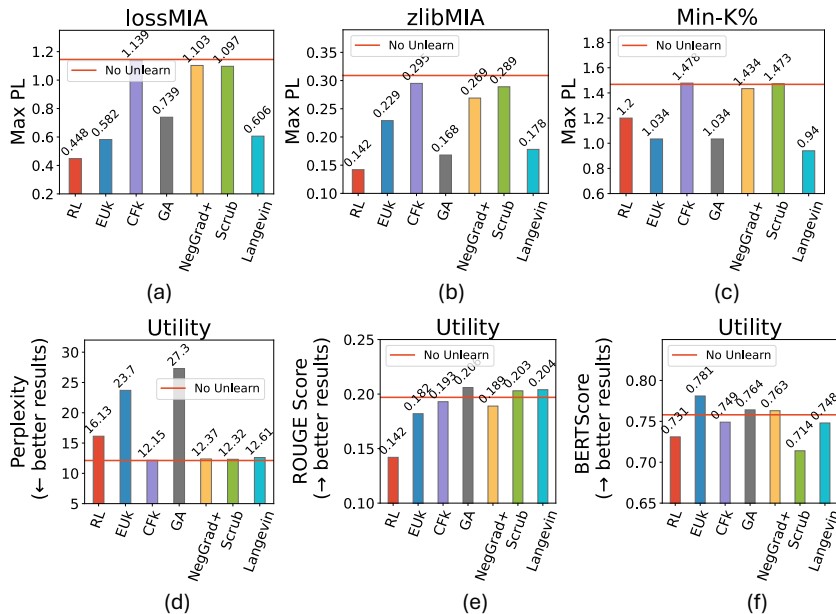

Figure 15: Benchmarking unlearning approaches via our minority-aware evaluation for GPT-2 on Enron-email dataset. (a)-(c): Maximum privacy leakage (PL) over three cases (`Random`, `Canary`, and `Minority`) for lossMIA, zlibMIA, and Min-K% attacks respectively. (d-f): Worst utility performance over the three cases of each method.

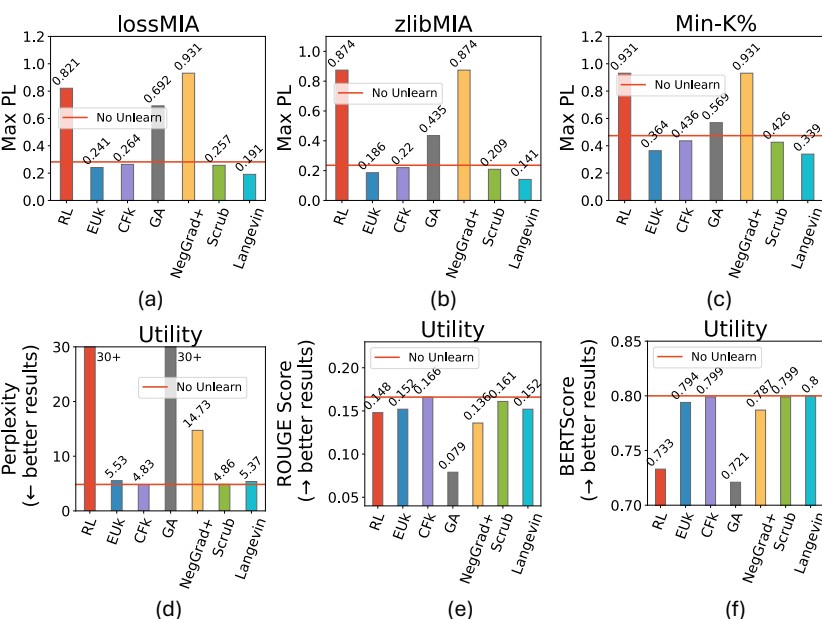

Figure 16: Benchmarking unlearning approaches via our minority-aware evaluation for Llama-2 on Enron-email dataset. (a)-(c): Maximum privacy leakage (PL) over three cases (`Random`, `Canary`, and `Minority`) for lossMIA, zlibMIA, and Min-K% attacks respectively. (d-f): Worst utility performance over the three cases of each method.

† **Experiments to Q2.** In Fig. 17, we report the degree of largest underestimation in privacy leakage compared to Random settings across different complexity units under Min-k% attacker. Detailed results under each settings are reported in Table 19 and 20.

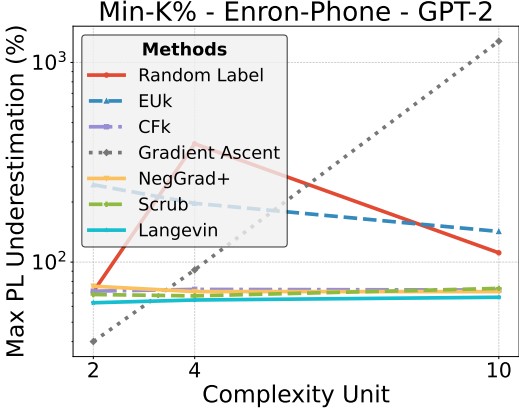

Figure 17: Degree of the largest underestimation in privacy leakage (Canary, Minority) compared to Random settings across varying Complexity Units.

Table 19: PL Scores for MIA across Three Settings for GPT-2 on Enron (Phone) under Different Complexity Units

| | Enron-Phone GPT-2 Min-k% | | | | | | | | |
|---|---|---|---|---|---|---|---|---|---|
| Methods | Complexity Units 2 | | | Complexity Units 4 | | | Complexity Units 10 | | |
| | Random | Canary | Minority | Random | Canary | Minority | Random | Canary | Minority |
| Random Label | 0.166 | 0.284 | 0.136 | -0.059 | 0.037 | 0.290 | 0.168 | 0.320 | 0.355 |
| EUk | 0.068 | 0.169 | 0.234 | 0.068 | 0.174 | 0.202 | 0.092 | 0.216 | 0.223 |
| CFk | 0.302 | 0.445 | 0.518 | 0.300 | 0.438 | 0.519 | 0.298 | 0.435 | 0.514 |
| Gradient Ascent | -0.175 | -0.245 | -0.203 | -0.245 | -0.469 | -0.352 | -0.031 | -0.012 | -0.427 |
| NegGrad+ | 0.309 | 0.452 | 0.543 | 0.309 | 0.447 | 0.529 | 0.298 | 0.452 | 0.510 |
| Scrub | 0.311 | 0.443 | 0.525 | 0.313 | 0.447 | 0.525 | 0.306 | 0.445 | 0.532 |
| Langevin | 0.163 | 0.259 | 0.265 | 0.161 | 0.259 | 0.265 | 0.159 | 0.259 | 0.265 |

Table 20: Perplexity for MIA across Three Settings for GPT-2 on Enron (Phone) under Different Complexity Units

| | Enron-Phone GPT-2 Perplexity | | | | | | | | |
|---|---|---|---|---|---|---|---|---|---|
| Methods | Complexity Units 2 | | | Complexity Units 4 | | | Complexity Units 10 | | |
| | Random | Canary | Minority | Random | Canary | Minority | Random | Canary | Minority |
| Random Label | 29.85 | 28.93 | 30+ | 30+ | 30+ | 30+ | 30+ | 30+ | 30+ |
| EUk | 30+ | 30+ | 30+ | 30+ | 30+ | 30+ | 23.64 | 23.65 | 26.60 |
| CFk | 12.72 | 12.71 | 12.72 | 12.72 | 12.72 | 12.73 | 12.67 | 12.67 | 12.67 |
| Gradient Ascent | 16.63 | 15.76 | 29.28 | 30+ | 30+ | 30+ | 30+ | 30+ | 30+ |
| NegGrad+ | 12.86 | 12.87 | 12.83 | 12.84 | 12.86 | 12.88 | 12.83 | 12.80 | 12.88 |
| Scrub | 12.94 | 12.95 | 12.96 | 13.11 | 13.09 | 13.19 | 13.09 | 12.88 | 12.96 |
| Langevin | 13.84 | 13.85 | 13.88 | 13.85 | 13.86 | 13.88 | 13.84 | 13.88 | 13.88 |

