# OpenReview forum: "Underestimated Privacy Risks for Minority Populations in Large Language Model Unlearning"
_ICLR.cc/2025/Conference — Submitted to ICLR 2025_

### Official Review · Reviewer_idfe · 2024-10-31

**Soundness:** 3
**Presentation:** 3
**Contribution:** 2
**Rating:** 5
**Confidence:** 4

**Summary:**

This paper studies the evaluation pipeline of LLM unlearning (i.e., efficient LLM modification so that it becomes statistically indistinguishable from a model retrained from scratch, without the data subject to removal), and identifies a flaw in the existing evaluation approaches.

**Strengths:**

1. This paper highlights an issue with LLM unlearning evaluation, being data dependent. The current evaluation chooses to forget the dataset uniformly at random.
2. This paper shows that minority groups experience at least 20% more privacy leakage in most cases across combinations of six unlearning approaches.
3. This paper calls for a more careful LLM unlearning efficacy evaluation.

**Weaknesses:**

Given the "right to be forgotten" and too expensive cost of re-training LLMs from scratch without the data subject to removal, machine unlearning techniques have been proposed.


As there is no formal unlearning guarantee for deep neural networks and LLMs, evaluation of LLM unlearning uses membership inference attacks as follows:
- Randomly select data for removal from the training set to create a "forget" dataset,
- Apply unlearning techniques to the LLM
- Use membership inference attacks to compare the unlearned models against models retrained without the removed data.

This paper identified a flaw in this evaluation pipeline (mainly in the first step): "the unlearning privacy risk of minority populations within the training set is severely underestimated since the minority data are less likely to be selected in the unlearning evaluation pipeline."


This paper proposes the need of using worst-case data for creating forget dataset in the evaluation pipeline. However, the suggested choice of worst-case scenario suffers from multiple issues:

1. This paper assumes that the worst case data corresponds to minority data.
2. This paper defines minority data based on only personally identifiable information.
3. This paper creates canaries by replacing personally identifiable information in randomly chosen data within forget dataset with least frequent ones.



This paper proposes a minority-aware LLM unlearning evaluation protocol to address the limitation of existing evaluation pipelines. However, the proposal depends heavily on the above choices: worst-case data, minority definition and canary creations. In addition, the broader motivation is unclear- for example if the data subject to removal does not blog to least frequent data, it is not clear what the applicability of the proposal would be.


The main finding of this paper (privacy risks of minority groups in the training data are usually underestimated) seems related to several existing works studying disparate vulnerability against MIAs.


Minors:
- All text in lines 197-215 + a table + a figure aim to say that frequencies of items vary in a dataset, or I am missing something non-trivial?

- "In the provided example,  Dforget unlikely contains the least frequent email with the 484 area code. As a result, the corresponding unlearning privacy evaluation may underestimate the privacy risk of minorities if unlearning minority data is inherently more challenging."" --> Not clear how the latter got concluded just because of being least frequent?

- bold claims without supports:
	- 237-238: "albeit a similar idea extends beyond PIIs"
	- 139-140: "albeit our methodology extends to other cases whenever the indistinguishability to the retrained model is an appropriate metric"




- 93-94: "the right to be forgotten should be respected for all individuals" --> how to define individual? each individual has one record or multiple records? This is not discussed later on in the paper.

- 40-41: missing refs for GDPR and right to be forgotten

- 262: "it cannot quantify underestimate the privacy risk for minorities in the real-world setting" --> grammar issue


- 377: "Enron(Klimt & Yang, 2004) and ECHR(Chalkidis et al., 2019)." --> missing space

**Questions:**

1. Why minority data should be considered as worst-case data for LLM unlearning evaluations?
2. Why minority data should be defined based on personally identifiable information? How generalisable the results/findings are when using alternative minority definitions?
3. Why is the suggested way of creating canaries relevant?
4. Why do the chosen minorities suffer from at least 20% more privacy leakage in studied cases? Is it due to being least frequent or their context/semantics?
5. How are these results compared to those works analyzing disparate vulnerability against MIAs such as Kulynych et. al., Disparate Vulnerability to Membership Inference Aacks (https://arxiv.org/pdf/1906.00389).

---

> ### Author Response · Authors · 2024-11-21
>
> We are deeply grateful to Reviewer idfe for the comprehensive feedback. Here, we will address these questions and concerns.
>
> >W1 & Q1 “This paper assumes that the worst case data corresponds to minority data.” “Why should minority data be considered as worst-case data for LLM unlearning evaluations?”
>
> We sincerely thank Reviewer idfe for raising this concern.  We acknowledge that the language we used in the submission could result in a potential misunderstanding regarding our claim. Here, we first will clarify our claim, and then will specify how we have modified the manuscript to clarify this.
>
> The core premise of our paper is that the "right to be forgotten" applies universally (i.e. to every individual) necessitating unlearning methods to be ideally evaluated under the worst-case scenario.  This underscores the need to move beyond existing evaluation pipelines, which focus on average-case scenarios, and instead develop frameworks that assess performance for individuals or groups disproportionately impacted by unlearning methods.
>
> To clarify, we do not claim that our minority-aware framework necessarily provides worst-case scenario results exactly. A minority group may or may not correspond with the worst case scenario, but most likely will be closer to it compared to the existing evaluation pipelines that are inherently focusing on average-case scenarios. Prior work has shown that minority data tend to have stronger memorization effects [1], potentially making them more challenging to unlearn. While a worst case analysis may be too difficult or overly restrictive in some practical scenarios, it is still crucial to ensure that unlearning evaluations account for the performance on under-represented groups. Our paper takes a first step in this direction by leveraging minority group samples—often particularly sensitive in real-world applications—as identifiers to illustrate the limitations of standard evaluation methods. Our findings demonstrate that unlearning methods perform significantly worse for minority groups compared to average-case scenarios. While our framework does not necessarily quantify worst-case risks exactly, it provides a better estimate on such scenarios.
>
> In light of this valuable feedback, we have updated the manuscript such that we only mention the need for worst-case analysis in the Introduction section to motivate the need to go beyond average case approaches, and refrain from using this language in the methodology and the rest of the paper to avoid any confusion. We also make it clear that while our work is motivated by the need for considering the worst case scenario (as opposed to the average case), it only focuses on minority subgroups, as they are more prone to have higher privacy leakage risks.
>
> [1] Carlini et al. Quantifying memorization across neural language models. In ICLR 2023.
>
>
> >W2 & Q2 “This paper defines minority data based on only personally identifiable information.” “Why should minority data be defined based on personally identifiable information? How generalisable are the results/findings when using alternative minority definitions?”
>
> Thank Reviewer idfe for raising this concern. We chose personally identifiable information (PII) as the identifier for our analysis because PII is highly sensitive, often the subject of user unlearning requests, and a central focus in privacy and unlearning research [2,3,4]. This choice aligns with real-world scenarios where unlearning sensitive data is both practical and necessary. Our minority-aware framework, however, is generic and  generalizes to other identifiers, including minority groups defined based on copyrighted information or specific text corpora. Exploring additional identifiers in future work could further expand the applicability and coverage of our study.
>
> [2] Jang et al. Knowledge unlearning for mitigating privacy risks in language models. In ACL 2023.
>
> [3] Lukas et al. Analyzing leakage of personally identifiable information in language models. In 2023 IEEE Symposium on Security and Privacy.
>
> [4] Nasr et al. Scalable extraction of training data from (production) language models. arXiv preprint.

---

> > ### Comment · Reviewer_idfe · 2024-11-22
> > **Worst case scenario**
> >
> > Thanks for the response.
> >
> > > The core premise of our paper is that the "right to be forgotten" applies universally (i.e. to every individual) necessitating unlearning methods to be ideally evaluated under the worst-case scenario.
> >
> > Given this, I would expect to see an approach for finding the worst-case data for unlearning. One suggestion that you might find helpful is to look into those works that study the privacy protection of defences on the most vulnerable data points.  For example, [1] shows that "privacy is not an average-case metric!", which is related to your work/message.
> >
> > [1] Michael Aerni, Jie Zhang, and Florian Tramèr. "Evaluations of Machine Learning Privacy Defenses are Misleading." arXiv preprint arXiv:2404.17399 (2024).
> >
> > [2] Nicholas Carlini, Chang Liu, Úlfar Erlingsson, Jernej Kos, and Dawn Song.  The secret sharer: Evaluating and testing unintended memorization in neural networks. In USENIX Security Symposium.
> >
> > [3] Matthew Jagielski, Jonathan Ullman, and Alina Oprea. Auditing differentially private machine learning: How private is private sgd? Advances in Neural Information Processing Systems (2020)

---

> > > ### Author Response · Authors · 2024-11-22
> > >
> > > We sincerely thank the reviewer for their thoughtful feedback. Our response to this comment has three layers:
> > >
> > > **First**, the works [1-3] cited by the reviewer have been discussed in our paper as evidence of the importance of moving beyond “average-case” analysis. While these works aim to construct worst-case scenarios, they do not guarantee that the identified samples truly represent the worst cases. In this sense, these studies align with ours: they focus on identifying samples that are more vulnerable to privacy attacks in specific ways, without claiming definitive identification of worst-case samples.
> > >
> > > **Second**, a typical approach to privacy auditing relies on *sample-specific MIAs*, which require training *thousands of shadow models for each point* (e.g., [1]). Additionally, for unlearning evaluations, it is necessary to account for the worst-case data points for *each unlearning method*, which would require examining every forget point. However, these approaches are not feasible for large language models due to their computational complexity. Moreover, identifying worst-case samples in text data is likely more challenging than in the modalities explored in these works. As we discuss in our response to Q4, our experiments suggest that the frequency of subgroup samples significantly influences unlearning performance. However, other factors—such as the content of private information, context, semantics, and unexplored dimensions—may also contribute to the disparate vulnerabilities of certain samples. Without a comprehensive understanding of these factors, identifying worst-case samples a priori remains unattainable. Our work takes an important first step by focusing on one specific factor: the frequency of samples within a subgroup.
> > >
> > > **Finally**, while studying worst-case scenarios is critical, we argue that quantifying the disparate impact of unlearning algorithms on minority groups is highly valuable in its own right. This perspective aligns with previous works in the privacy literature, such as Kulynych et al. [4], which investigates the disparate vulnerability of minority subgroups to privacy attacks. We emphasize that while the broader motivation of our work is driven by the principle that “the right to be forgotten” should be applied universally, our proposed framework provides valuable insights that go beyond worst-case analysis. For example, subgroup-level metrics are essential for ensuring fairness in unlearning algorithms, a perspective not necessarily captured by worst-case analysis alone.
> > >
> > >
> > > [1] Michael Aerni, Jie Zhang, and Florian Tramèr. "Evaluations of Machine Learning Privacy Defenses are Misleading." arXiv preprint arXiv:2404.17399 (2024).
> > >
> > >
> > > [2] Nicholas Carlini, Chang Liu, Úlfar Erlingsson, Jernej Kos, and Dawn Song. The secret sharer: Evaluating and testing unintended memorization in neural networks. In USENIX Security Symposium.
> > >
> > >
> > > [3] Matthew Jagielski, Jonathan Ullman, and Alina Oprea. Auditing differentially private machine learning: How private is private sgd? Advances in Neural Information Processing Systems (2020).
> > >
> > >
> > > [4] Kulynych et al. Disparate vulnerability to membership inference attacks. In PET 2022.

---

> ### Author Response · Authors · 2024-11-21
>
> >W3 & Q3 “This paper creates canaries by replacing personally identifiable information in randomly chosen data within the forget dataset with least frequent ones.” “Why is the suggested way of creating canaries relevant?”
>
> The Canary setting in our study serves as a controlled baseline, enabling us to control variables for more precise conclusions. In our experiments, we used highly sensitive personally identifiable information (PII) as identifiers to examine privacy leakage across random and minority settings based on frequency. However, since the non-PII text in each sample varies between these settings, we introduced the Canary setting to specifically assess how the population size of identifiers alone influences privacy leakage. This approach ensures that our evaluation isolates the effects of identifier frequency on privacy risks, making the Canary setting a relevant and essential part of understanding privacy leakage in minority-aware evaluations.
>
> >W4 “In addition, the broader motivation is unclear- for example if the data subject to removal does not blog to least frequent data, it is not clear what the applicability of the proposal would be.”
>
> We appreciate the concern raised by Reviewer idfe. However, the right to be forgotten is universal, meaning that every user can submit an unlearning request and everyone has the right to be forgotten. Consequently, privacy protection and the unlearning process must aim to ensure protection or removal for ***every data point***. Furthermore, unlearning methods should ideally be evaluated under “worst-case” scenarios to ensure robust privacy guarantees. When selecting the forget set to compare different unlearning methods, it is therefore crucial to include points that closely mimic “worst-case” scenarios. In our study, we take the first step towards the above objective by studying the minority subgroups, which have been shown to exhibit stronger memorization effects and are more likely to be challenging to unlearn.
>
> >W5 Minors: “All text in lines 197-215 + a table + a figure aim to say that frequencies of items vary in a dataset, or I am missing something non-trivial?”
>
> Our intention in this section is to convey the following points: (1) Provide a brief introduction to the Enron dataset, highlighting its sensitive sample information. (2) Help readers better understand examples of PII and the population distribution within the dataset. (3) Emphasize the counts as they are crucial for selecting the forget set size and ensuring that the crafted samples in the canary set remain representative of a minority group within the population.
>
> >W5 Minors: “In the provided example, Dforget unlikely contains the least frequent email with the 484 area code. As a result, the corresponding unlearning privacy evaluation may underestimate the privacy risk of minorities if unlearning minority data is inherently more challenging." --> Not clear how the latter got concluded just because of being least frequent?”
>
> Thank you for pointing this out. What we aim to convey is that if $D_\text{forget}$ is selected through uniform random sampling, it is unlikely to include minority data due to its lower frequency. Consequently, if unlearning minority data is inherently more challenging and leads to greater privacy leakage, the existing evaluation pipeline would underestimate privacy risks.
>
> To address the potential confusion caused by this statement, we have revised the corresponding description in the text for greater clarity.
>
> >W5 Minors: bold claims without supports: 1. 237-238: "albeit a similar idea extends beyond PIIs"  2. 139-140: "albeit our methodology extends to other cases whenever the indistinguishability to the retrained model is an appropriate metric"
>
> For the first point, our study focuses on using sensitive PIIs as identifiers to study population groups, as these are a significant privacy concern in real-world scenarios. Nevertheless, the idea that privacy is a worst-case metric holds regardless of the identifier. We agree that it is an interesting future direction to conduct a comprehensive study for different identifiers, such as copyrighted information or specific text corpora.
>
> For the second point, we argue that our identification of a flaw in the standard LLM unlearning evaluation pipeline is independent of the specific privacy metric used. Data points exhibit varying levels of memorization effects, and the difficulty of unlearning certain points may vary accordingly. Consequently, a “worst-case” scenario inherently exists, and the limitations of traditional evaluation methods based on uniform sampling are likely to underestimate privacy risks regardless of the metric chosen.

---

> ### Author Response · Authors · 2024-11-21
>
> >W5 Minors: 93-94: "the right to be forgotten should be respected for all individuals" --> how to define individual? each individual has one record or multiple records? This is not discussed later on in the paper.
>
> We thank the Reviewer idfe for pointing out a detail that may cause confusion in our paper. In the literature on unlearning methods, the unlearning task is typically categorized into instance-level unlearning (where each individual corresponds to a single record) and entity-level unlearning (where each individual corresponds to multiple records) [5]. The former involves record-based unlearning, where each record is treated as an independent sample, while the latter focuses on unlearning all records associated with a specific entity. Our study focuses on instance-level unlearning, where each record is independently targeted for removal, which is a common setting in privacy research. Our approach could potentially extend to multiple records, but this is beyond the scope of the current paper. To ensure clarity, we have emphasized this point in the revised version of the paper (Experimental Section).
>
> [5] Maini et al. Tofu: A task of fictitious unlearning for llms. In COLM 2024.
>
> >W5 Minors: 40-41: missing refs for GDPR and right to be forgotten 262: "it cannot quantify underestimate the privacy risk for minorities in the real-world setting" --> grammar issue 377: "Enron(Klimt & Yang, 2004) and ECHR(Chalkidis et al., 2019)." --> missing space
>
> We thank Reviewer idfe for pointing out these typos. We have corrected them in the revised version of the paper.
>
> >Q4 “Why do the chosen minorities suffer from at least 20% more privacy leakage in studied cases? Is it due to being least frequent or their context/semantics?”
>
> This is a great question. As for whether this disparity is due to lower frequency vs. context/semantics, our results indicate that low frequency plays an important role here. Specifically, our Canary setting controls for any changes in context/semantics in the data (it only replaces PII with least frequent PII, and preserves the rest of the text in the sample). Our controlled Canary vs. Random experiments confirm greater privacy leakage for data that includes information from minority subgroups, indicating that the lower frequency of minority data could be the main culprit. Further, it has been shown that large models tend to exhibit stronger memorization effects on data from under-represented subgroups [6]. This suggests that such data may also be more challenging to unlearn in such models and resulting in higher privacy leakage.
>
> We believe that providing a scientific response to this question is both highly important and of significant interest.  In this work, however, our focus is on demonstrating the existence of this gap and presenting a framework for measuring it. Investigating the underlying reasons for why this gap exists would constitute a separate and valuable contribution to the unlearning literature. As such, we consider it beyond the scope of this paper and have left it for future work.
>
> [6] Carlini et al. Quantifying memorization across neural language models. In ICLR 2023.
>
>
> >Q5 “How are these results compared to those works analyzing disparate vulnerability against MIAs such as Kulynych et. al., Disparate Vulnerability to Membership Inference Attacks”
>
> Thank Reviewer idfe for the question. The paper by Kulynych et al. focuses on disparate vulnerabilities to MIAs by analyzing the distributional generalization gap, demonstrating these vulnerabilities on logistic regression and two-layer ReLU neural networks. However, the MIA scenario we consider is distinct from the standard MIA hypothesis testing framework fundamentally. In the standard setting, MIAs assess whether a given sample is part of the training set by comparing models trained on adjacent datasets (with or without the sample). In contrast, in the context of LLM unlearning, MIAs are employed to evaluate the indistinguishability between unlearned and retrained models—measuring whether a sample originates from the unlearned model or the retrained model, where the unlearned model includes both pre-training/fine-tuning and unlearning processes. This shift in focus fundamentally alters the nature of MIA attacks, making them dependent on removal requests. Our work specifically calls attention to evaluating removal requests for under-represented groups, an aspect that is often overlooked in current LLM unlearning literature.
>
> Further, while previous privacy studies, including Kulynych et al., have discussed minority group vulnerabilities, we are the first to identify the critical issues within the unlearning evaluation pipeline. Specifically, our work highlights how the traditional pipeline systematically underestimates privacy risks, particularly in the unlearning context. This represents a significant contribution to advancing the evaluation of unlearning methods in LLMs.

---

### Official Review · Reviewer_hsY8 · 2024-11-02

**Soundness:** 3
**Presentation:** 2
**Contribution:** 2
**Rating:** 6
**Confidence:** 2

**Summary:**

This paper conducts a benchmark evaluation of different machine unlearning approaches, and computes membership inference metrics on the unlearned models.  They find that prior work underestimates the privacy risk on minority groups by about 20% in the settings they studied.  Their findings are fairly robust across a variety of different unlearning methods, MIA metrics, base models, datasets, and other experimental settings.   Based on this finding, the authors advocate for a minority-aware evaluation, and discuss which methods perform best under this evaluation.

**Strengths:**

* Paper introduces a solid benchmark and a carefully crafted set of experiments.
* This main findings are interesting, and could help shape future research in this space.

**Weaknesses:**

* The main finding that MIA attacks have different success rates on different subsets of the population is not surprising, and is in line with other work in the privacy + fairness space.
* A good number of experiments were done, but only two main findings came out from them: (1) that MIA metrics differ based on { Random, Canary /Minority } and (2) that  Langevin Unlearning provided the best balance between privacy and utility.  Are there more findings or observations you can make based on the data you collected?  You should target at least one key takeaway for each Figure/Table you have.

**Questions:**

* What are the limitations of your methodology?
* Can you offer any insight into why Langevin Unlearning does better under a minority-aware evaluation?
* Table 4 and 5 are pretty busy, and there is no reference to Table 4 in the text.  If you can help the reader by telling them what they should look at in the table and how to interpret the full results, that would be nice.

---

> ### Author Response · Authors · 2024-11-21
>
> We greatly thank Reviewer hsY8 for recognizing the comprehensiveness of our LLM unlearning evaluation and for acknowledging the interest of our findings. Below, we address the questions and concerns raised by Reviewer hsY8.
>
> >W1 “The main finding that MIA attacks have different success rates on different subsets of the population is not surprising, and is in line with other work in the privacy + fairness space.”
>
> We greatly thank Reviewer hsY8 for the thoughtful point. However, we would like to emphasize that our work is the first to identify a critical flaw in the current LLM unlearning evaluation pipeline, showing that privacy leakage is substantially underestimated under the existing framework. This insight highlights the need for more comprehensive unlearning evaluations, as uniform sampling alone is insufficient. While some privacy literature aligns with a similar principle, our work highlights this flaw within the context of unlearning evaluation, which is crucial and has never been identified and addressed in the previous literature.
>
> >W2 “A good number of experiments were done, but only two main findings came out from them: (1) that MIA metrics differ based on { Random, Canary /Minority } and (2) that Langevin Unlearning provided the best balance between privacy and utility. Are there more findings or observations you can make based on the data you collected? You should target at least one key takeaway for each Figure/Table you have.”
>
> To highlight the flaw in the current unlearning evaluation pipeline and demonstrate that it is universal, our study incorporates diverse models, multiple PII-based datasets, and various MIA attackers to comprehensively verify our claims. Different experimental setups will improve the validity and persuasiveness of our findings. Beyond the two main takeaways mentioned, we also provide additional insights in the ablation study and appendix. Specifically, we analyze the privacy-utility trade-off curves of stable methods (SCRUB and Langevin Unlearning) under different hyperparameter settings (Revised Version Fig.6, Appendix Fig.14). Furthermore, we explore the behavior of different unlearning methods under varying unlearning epochs and forget set sizes (Fig.7), offering a more detailed understanding of their performance across scenarios.
>
> >Q1 “What are the limitations of your methodology?”
>
> We appreciate Reviewer hsY8’s thoughtful question. The ultimate goal of privacy evaluation in LLM unlearning is to quantify the effectiveness of different unlearning methods under “worst-case” scenarios. Minorities have been shown to exhibit stronger memorization effects and are, therefore, likely to be harder to unlearn. Our work takes a first step toward addressing "worst-case" privacy leakage by focusing on minority group identifiers. However, we acknowledge that fully identifying and addressing the true worst-case samples remains a challenging task that requires further exploration.
>
> >Q2 “Can you offer any insight into why Langevin Unlearning does better under a minority-aware evaluation?”
>
> Thank Reviewer hsY8 for the question. We believe that Langevin Unlearning’s superior performance under a minority-aware evaluation can be explained from a privacy perspective. In traditional privacy-preserving methods, injecting (Gaussian) noise is a standard approach to safeguard privacy. For instance, noisy training can provide certain theoretical privacy guarantees in the differential privacy (DP) framework. It is known to the privacy literature [1,2] that DP bounds can be used to bound the MIA vulnerability of individual samples or subgroups. While their settings differ from our unlearning scenario, the results may shed some light on the effectiveness of Langevin Unlearning, which is the only noise-based tested unlearning approach. To date, all approximate unlearning approaches with theoretical guarantees are all noise-based approaches, albeit most of the works are tailored for strongly convex problems [3,4,5]. For additional theoretical properties of Langevin Unlearning, please refer to [6].
>
> [1] Yeom et al. Privacy risk in machine learning: Analyzing the connection to overfitting. In 2018 IEEE 31st computer security foundations symposium (CSF).
>
> [2] Kulynych et al. Disparate vulnerability to membership inference attacks. In PET 2022.
>
> [3] Sekhari et al. Remember what you want to forget: Algorithms for machine unlearning. In NeurIPS 2021.
>
> [4] Ullah et al. From adaptive query release to machine unlearning. In ICML 2023.
>
> [5] Guo et al. Certified data removal from machine learning models. In ICML 2020.
>
> [6] Chien et al. Langevin Unlearning: A New Perspective of Noisy Gradient Descent for Machine Unlearning. In NeurIPS 2024.

---

> ### Author Response · Authors · 2024-11-21
>
> >Q3 “Table 4 and 5 are pretty busy, and there is no reference to Table 4 in the text. If you can help the reader by telling them what they should look at in the table and how to interpret the full results, that would be nice.”
>
> We would like to thank Reviewer hsY8 for the helpful suggestion, and have made adjustments to the manuscript accordingly. We hope these modifications clarify the following.  In Section 6.1, to verify our claims, we evaluate unlearning methods across different models (GPT-2 and LLaMA-2) and sensitive datasets , which we report in Tables 2-5. In each table, we highlight the extent to which privacy leakage is underestimated in Canary and Minority scenarios compared to the Random setting, under different attacker setups. We also emphasize in Section 6.1 that the results across the four tables are consistent, demonstrating that privacy risks are underestimated under the current evaluation framework.

---

> > ### Comment · Reviewer_hsY8 · 2024-11-26
> >
> > Thanks for the response to my comments, I think there are some nice ideas and findings in your paper.

---

### Official Review · Reviewer_qsaJ · 2024-11-04

**Soundness:** 3
**Presentation:** 3
**Contribution:** 3
**Rating:** 6
**Confidence:** 2

**Summary:**

The common method of evaluating the privacy leakage of unlearning in large language models is based on a chosen dataset $D_{forget}$ to unlearn from that model, say, $M^{(1)}$. The paper argues that a current method of using a random $D_{forget}$ significantly underestimates the real privacy leakage of the unlearned model. To do this, the paper proposes the method of analyzing the privacy leak of a synthetic dataset $D_{canary}$ created by replacing personally identifying information from $D_{forget}$ with the most unlikely personally identifying information from the whole training set $D_{train} = D_{keep} \cup D_{forget}$. The paper compares the privacy leakage of the $M^{(1)}$ and $M^{(2)}$ that were trained on $D_{forget} \cup D_{keep}$ and $D_{canary} \cup D_{keep}$ respectively and then un-trained on $D_{forget}$ and $D_{canary}$ respectively. Differences in audits in the unlearned models and a model retrained on $D_{keep}$ from scratch shows privacy leakage. Since a synthetic dataset might not represent real world scenarios, the paper does a similar analysis where $D_{forget}$ is taken from the most unlikely data in $D_{train}$.

**Strengths:**

* The paper highlights some weakness in current privacy evaluations for algorithm-agnostic unlearning techniques in LLMs and provides methods for improving privacy audits for these LLMs.

* The paper covers multiple unlearning algorithms and multiple membership inference attacks.

* The paper overall is clearly written, for example it has clear explanations of the background such as model unlearning.  Figure 1 is a clear overview of the pipeline proposed in the paper.

**Weaknesses:**

* The paper seeks to explain that their evaluations of MIAs based on how *Canaries* and *Minorities* gives higher *PrivLeak* scores compared to *Random*. However, there is no justification for why the *PrivLeak* score is used to begin with. This seems important particularly given that the subsequent analysis in the paper hinges on this choice.

* The paper analyses the privacy-utility trade-off of models after unlearning by comparing the *PrivLeak* score and LLM perplexity. The analysis could benefit from other utility measures (for example it doesn't capture semantic meaning, and LLMs could be very confident about an incorrect prediction).

**Questions:**

* Why is it useful to study the *Canaries* set if synthetic data does not reflect real world scenarios?

* What is the justification for 10 unlearning units? How does the large underestimation in PL compare to *random* under different unlearning budgets?
  * For an LLM would it be practical to allow for longer unlearning times especially considering the retraining cost is much larger in comparison?
* The paper highlights limitations in the typical evaluation pipeline (*random*). Are there limitations in this new pipeline?

---

> ### Author Response · Authors · 2024-11-21
>
> We sincerely thank Reviewer qsaJ for acknowledging the comprehensiveness of our paper in supporting the core message that “the current LLM unlearning evaluation pipeline can significantly underestimate privacy leakage”. Below, we address the questions and concerns raised by Reviewer qsaJ.
>
> >W1 “The paper seeks to explain that their evaluations of MIAs are based on how Canaries and Minorities give higher PrivLeak scores compared to Random. However, there is no justification for why the PrivLeak score is used to begin with. This seems important particularly given that the subsequent analysis in the paper hinges on this choice.”
>
> Thank you for this insightful question. Our claim about identifying a flaw in the standard LLM unlearning evaluation pipeline is independent of the specific privacy metric used. Data points exhibit varying levels of memorization effects, and the difficulty of unlearning specific points may differ. As a result, a “worst-case” scenario will always exist, and the shortcomings of traditional evaluation methods on uniform sampling will likely underestimate privacy regardless of the chosen metric. We selected PrivLeak for this study for two main reasons: (1) existing research commonly uses MIAs to quantify privacy leakage [1,2], and PrivLeak captures the normalized difference in MIAs between unlearned and retrained models, making it an effective indicator of privacy leakage. (2) Additionally, the benchmark paper MUSE also employs and thoroughly explains PrivLeak, allowing for a direct comparison with prior work. We have refined the relevant descriptions in the paper to enhance clarity.
>
> [1] Thudi et al. Unrolling sgd: Understanding factors influencing machine unlearning. In 2022 IEEE 7th European Symposium on Security and Privacy.
>
> [2] Liu et al. Model sparsity can simplify machine unlearning. In NeurIPS 2023.
>
> >W2 “... The analysis could benefit from other utility measures (for example it doesn't capture semantic meaning, and LLMs could be very confident about an incorrect prediction).”
>
> We thank the reviewer for this valuable suggestion. LLM perplexity was chosen as the utility metric because it is widely recognized as a standard evaluation measure for LLMs, and prior studies on privacy and unlearning in LLMs have also primarily relied on perplexity as their main utility metric [3,4,5]. Nevertheless, we agree with reviewer qsaJ that our work can be further strengthened by examining different utility metric that captures LLM semantic meaning. As a result, we additionally report BERTScore [6], ROUGE Score [7] as other utility metrics in Appendix Sec.D Fig.15-16. The results are consistent, showing that Langevin Unlearning achieves the best privacy-utility trade-off, while Gradient Ascent and Random Label methods remain unstable.
>
> [3] Lukas et al. Analyzing leakage of personally identifiable information in language models. In 2023 IEEE Symposium on Security and Privacy.
>
> [4] Zhang et al. Composing parameter-efficient modules with arithmetic operation. In NeurIPS 2023.
>
> [5] Ilharco et al. Editing models with task arithmetic. In ICLR 2023.
>
> [6] Zhang et al. Bertscore: Evaluating text generation with bert. In ICLR 2020.
>
> [7] Lin et al. Rouge: A package for automatic evaluation of summaries. In Text summarization branches out.
>
> >Q1 “Why is it useful to study the Canaries set if synthetic data does not reflect real world scenarios?”
>
> While the Canary setting is synthetic, it serves as a controlled baseline in our study, allowing us to isolate variables effectively. In our experiments, we used personal identifiable information (PII) as identifiers to compare the privacy leakage across random and minority settings based on frequency. However, in real data, the samples differ not only in the PII but also in the remaining text. To control for the impact of this additional text (may lead to different semantic meanings), we included the Canary setting to specifically examine how the population size of identifiers alone affects privacy leakage.

---

> ### Author Response · Authors · 2024-11-21
>
> >Q2 “What is the justification for 10 unlearning units? How does the large underestimation in PL compare to random under different unlearning budgets? For an LLM would it be practical to allow for longer unlearning times especially considering the retraining cost is much larger in comparison?”
>
> The selection of 10 unlearning units reflects practical time constraints for unlearning methods in real-world applications. In our paper, we set the forget set to approximately 1% of the fine-tuning training set size. Given that we use 10 unlearning units with 5 training epochs during fine-tuning, this corresponds to an unlearning time budget of about 2% of the fine-tuning phase, which is practical for large model fine-tuning. Prior work [8] estimates unlearning time as 4%-10% of fine-tuning time, but they also point out that real-world applications often require an even smaller budget. Our study focuses on such efficient unlearning settings.
>
> In Section 6.3 (Fig. 7, Left), we conducted ablation studies on different unlearning budgets. Additionally, we further report the degree of largest underestimation in privacy leakage compared to random sampling in Appendix Sec.D Fig.17 and Table 19-20 as suggested by the reviewer qsaJ. The results demonstrate that across all unlearning budgets, privacy leakage for minority groups is significantly underestimated, with the degree of underestimation exceeding 60% for almost all unlearning methods.
>
> [8] Pawelczyk et al. Machine unlearning fails to remove data poisoning attacks. arXiv preprint.
>
> >Q3 “The paper highlights limitations in the typical evaluation pipeline (random). Are there limitations in this new pipeline?”
>
> We appreciate the reviewer’s thoughtful question. The ultimate goal of privacy evaluation in LLM unlearning is to quantify the effectiveness of different unlearning methods under “worst-case” scenarios. Minorities have been shown to exhibit stronger memorization effects and are, therefore, likely to be harder to unlearn. Our work takes a first step toward addressing "worst-case" privacy leakage by focusing on minority group identifiers. However, we acknowledge that fully identifying and addressing the true worst-case samples remains a challenging task that requires further exploration.

---

> ### Author Response · Authors · 2024-12-01
>
> Dear Reviewer qsaJ,
>
> Thank you once again for your thoughtful comments on our paper. We wanted to kindly check if you had any further thoughts or feedback on our response and the revisions we’ve made. Please feel free to let us know if there’s anything else we can clarify. Thanks!

---

### Author Response · Authors · 2024-11-21
**Response Summary to Reviews**

We sincerely appreciate the valuable feedback and insightful comments from all our reviewers. We are especially grateful for the recognition of the comprehensiveness and robustness of our experiments in highlighting the weaknesses of current privacy evaluation pipelines in LLM unlearning. Key questions raised include the design of the Canary settings, limitations of the current pipeline, and the generalizability of our framework to other identifiers. Below, we are eager to provide further clarifications on these points in the detailed responses to each reviewer.

In addition, we have made revisions to the paper based on the reviewers' suggestions and conducted additional experiments to address their questions, as detailed in Appendix D (Rebuttal Section), and will be integrated into our next revision. We also included several new experiments to further support our findings:

>In Section 6.1, we focus on evaluating whether the standard use of the MIA approach for empirical evaluation of unlearning efficacy accurately measures privacy leakage. To this end, we unified the forget set size for Llama-2 7B to 1% of the training set (500 samples) and updated the results in Tables 4–5 and Appendix Table 8.

>In Section 6.2, we evaluate the performance of various unlearning methods within our minority-aware framework. To provide a more comprehensive evaluation, we extended the benchmarking of unlearning methods from GPT-2 to include Llama-2 7B, as presented in Figure 4 (Right).

>In Section 6.3, based on our previous benchmarking observations that only SCRUB and Langevin Unlearning demonstrated stable performance across different datasets and models, we conducted new experiments to further explore these methods. Specifically, we systematically performed a hyperparameter search and compared the privacy-utility trade-offs of SCRUB and Langevin Unlearning across multiple datasets (Figure 5). Additionally, we included the transition curves of hyperparameters in the appendix to provide readers with better guidance on selecting hyperparameters for these methods.

We hope these additions and clarifications address the reviewers’ concerns and provide a stronger foundation for our claims.

---

### Meta-Review · Area_Chair_nGZN · 2024-12-14

**Metareview:**

## Summary of Contributions

This paper study an unlearning setting where an LLM model is first train using a dataset $D_{keep} \cup D_{forget}$ and an unlearning algorithm is ran on the LLM to "forget" the $D_{forget}$ part. Previous work studies privacy leakage by setting $D_{forget}$ to be a random subset of the entire data and runs certain attack, e.g. Membership Inference Attack (MIA). In this work, the authors propose to set $D_{foget}$ to either be (i) from synthetic "canary" data or (ii) from minority data. They show that this approach leads to higher privacy leakage.

## Strengths
- The finding that the leakage is higher in the minority case is quite intriguing and should be included in future works when evaluating privacy leakage.

## Weaknesses
- The method proposed in this paper seems somewhat arbitrary and leads to several confusions:
  - **Synethtic vs Real Dataset**: It is not clear why "canary" data is considered if we are interested in real data only.
  - **Experimental Set-up**: There are several other experimental setups that should be compared against to conclusively arrive at the supposed finding of this paper.
    - *Majority Forget Set*: In the current set-up, it is entirely possible that we see higher privacy leakage due to *homogenuity* in the forget set. To dispel this, some experiment of the following form should be considered: set $D_{forget}$ to be a random subset of the *majority* set and compute the privacy leakage.
    - *Random Forget Set*: It is also unclear whether it is fair to compute MIA using different forget sets. Another experiment that should be considered (or at least discussed) is to use the Random set, but then measured the privacy leakage *separately* for majority and minority. This is arguably more "fair" since we are comparing the leakage of the two groups against the same unlearning set.
- The presentation can be improved: It is not clear how some of the figures / texts contribute to the main findings of the paper. Also, while it is mentioned that several algorithms (such as Langevin Unlearning and SCRUB) are more robust than others, no explanation is provided as to why this is the case.

## Recommendation

Due to the aforementioned weaknesses, this paper will benefit from another significant revision before being published. Thus, we recommend rejection.

**Additional Comments On Reviewer Discussion:**

There were quite a lot of discussions regarding the choice of the setting in this paper, including what does "worst case" privacy means and whether it equates to minority, how the minority is defined, and how privacy leakage is measured. Although some concerns are alleviated, there are still quite a few concerns left (as mentioned in the meta-review).

---

### Decision · Program_Chairs · 2025-01-22

Reject